# Learnability-Driven Knowledge Assimilation
# for Class-Incremental Semantic Segmentation

**Xinyue Zhang** [* 1]  **Xu Zou** [* 1]  **Wanjia Luo** [2]  **Yanjie Wang** [† 1]  **Jiahuan Zhou** [3]  **Sheng Zhong** [1]  **Luxin Yan** [1]

## Abstract

Class-incremental semantic segmentation learns new classes while retaining old ones without access to past data. Although existing methods alleviate catastrophic forgetting on old classes, new-class performance remains limited. We identify that the key bottleneck arises from *low-margin regions*, where the logit of the ground-truth class is close to that of the most competitive non-ground-truth class. Our theoretical analysis suggests that optimization in these regions is characterized by high second-order margin sensitivity and a small stability radius, making learning prone to class confusion. Based on the above analysis, we propose **L**earnability-**D**riven **K**nowledge **A**ssimilation (LDKA), which targets low-margin learning via three complementary optimization strategies: (i) Progressive Margin Learning continuously reallocates pixel-wise optimization budget in a threshold-free manner, shifting emphasis from high-margin to low-margin regions; (ii) Smooth Knowledge Distillation applies second-order sensitivity damping along the margin direction and perturbation stabilization to suppress high-frequency updates and increase the stability radius; (iii) Misclassification-Aware Decoupling measures inter-class confusion with a competition matrix and decouples highly competitive class representations. Experiments show that LDKA improves mIoU on new classes while preserving performance on old classes across 9 incremental protocols.

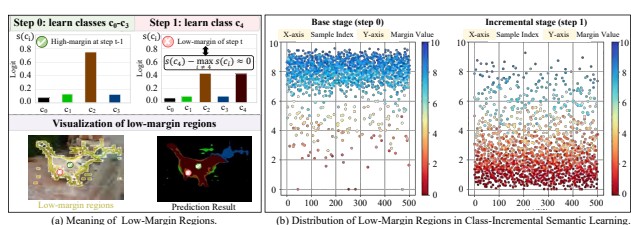

*Figure 1.* **Low-margin regions bottleneck CISS.** (a) Pixels with a small logit margin (e.g., top-1 $c_4$ vs. top-2 $c_2$) form *low-margin regions*, where predictions are uncertain and easily flipped. (b) After incremental learning, more samples are distributed in the low-margin region, resulting in class confusion.

## 1. Introduction

Class-incremental semantic segmentation (CISS) studies how to incrementally learn new classes over time when historical training data are not fully accessible due to privacy or storage constraints (Yang et al., 2023; Li & Hoiem, 2017). This is applied to fields such as autonomous driving and robot perception, where the key challenge is balancing *stability* for old classes and *plasticity* for new ones.

Recently, a dominant line of CISS has focused on stabilizing old-class representations through techniques such as knowledge distillation (Baek et al., 2022; Shang et al., 2023), contrastive learning (Zhao et al., 2023), replay (Kumari et al., 2022; Oh et al., 2022), and architectural expansion (Oh et al., 2022; Qin et al., 2021). Despite this progress, a persistent gap remains: new-class performance often shows minimal improvement as the incremental steps become longer. This limitation suggests that the bottleneck of CISS is not only explained by forgetting alone, but is also shaped by how optimization unfolds under strong stability constraints.

This paper reveals that the key bottleneck of CISS concentrates on the increasing distribution of samples in low-margin regions, which causes class confusion when learning new classes and leads to degraded performance (as shown in Fig. 1). Specifically, we use the *logit margin* to measure how ambiguous a pixel's prediction is, defined as the gap between the logit of the ground-truth class and the largest competing logit. When this gap is small, the pixel falls into the *low-margin region*. In this paper, we show that low-margin pixels are ill-conditioned for optimization. As we

---

[*]Equal contribution  [1] State Key Laboratory of Multispectral Information Intelligent Processing Technology, School of Artificial Intelligence and Automation, Huazhong University of Science and Technology, Wuhan, China [2]Department of Physics, Faculty of Science, National University of Singapore, Singapore [3]Wangxuan Institute of Computer Technology, Peking University, Beijing, China. Correspondence to: Yanjie Wang <aiawyj@hust.edu.cn>.

*Proceedings of the 43rd International Conference on Machine Learning*, Seoul, South Korea. PMLR 306, 2026. Copyright 2026 by the author(s).

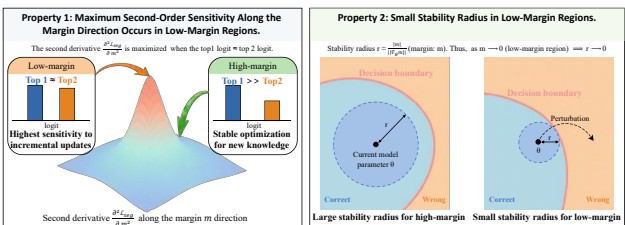

*Figure 2.* **Why low-margin regions bottleneck CISS.** Low-margin regions exhibit high second-order sensitivity along the margin direction and a small stability radius, making them highly sensitive to parameter updates and thus prone to mis-segmentation during incremental learning of new classes.

derive in Properties 1–2, a small logit margin induces high second-order sensitivity along the margin direction, and simultaneously yields a small stability radius (Fig. 2), so that even minor parameter updates or perturbations can readily flip predictions. Consequently, effective CISS demands explicitly targeting low-margin regions by reducing second-order sensitivity along the margin direction and addressing their small stability radius.

Motivated by the above analysis, we propose Learnability-Driven Knowledge Assimilation (LDKA) to tackle low-margin optimization through cohesive allocation-stabilization-decoupling optimization design: (i) Progressive Margin Learning (PML), which determines what to learn by progressively allocating the optimization budget, prioritizing high-margin regions early and gradually refining low-margin regions. (ii) Smooth Knowledge Distillation (SKD), which determines how to learn by stabilizing low-margin updates with directional second-order sensitivity damping and perturbation-based stabilization. (iii) Misclassification-Aware Decoupling (MAD), which determines where to separate by using a competition matrix to model misclassification and decoupling representations for highly confused class pairs. This work mainly makes three contributions:

- **Optimization diagnosis of low-margin regions in CISS.** We provide a diagnosis of low-margin regions in CISS, demonstrating that these regions exhibit high second-order sensitivity along the margin direction and a small stability radius, which leads to ill-conditioned optimization under incremental updates.

- **Learnability-Driven Knowledge Assimilation.** We propose **L**earnability-**D**riven **K**nowledge **A**ssimilation (LDKA), an optimization method that strengthens low-margin learning via: (i) *Progressive Margin Learning* with continuous, threshold-free, progress-aware allocation of optimization budget, (ii) *Smooth Knowledge Distillation* with directional second-order sensitivity damping and perturbation-based stabilization, and (iii)

*Misclassification-Aware Decoupling* that uses a competition signal to mitigate interference among classes.

- **Comprehensive empirical evaluation.** Experiments demonstrate that LDKA improves mIoU on newly emerging classes while preserving performance on old classes across 9 diverse incremental configurations.

## 2. Related Work

CISS aims to continuously learn new classes while preserving the segmentation performance of previously learned classes. Since training data from previous stages are often partially or entirely inaccessible due to privacy concerns or storage constraints, models trained in an incremental manner are prone to catastrophic forgetting. To address this challenge, existing CISS methods can be broadly categorized into three main paradigms: architecture expansion, replay-based methods, and regularization-based approaches.

**Architecture Expansion.** These approaches allocate additional branches or modules for different classes, thereby isolating old and new knowledge (Yoon et al., 2017; Qin et al., 2021). While such methods alleviate catastrophic forgetting to some extent, their model size and computational cost typically grow with the number of incremental steps.

**Replay.** Replay-based methods store a small set of old samples for rehearsal (Zhu et al., 2023; Cha et al., 2021), or generate pseudo-samples of old classes using generative models to approximate historical data distributions (Maracani et al., 2021; Chen et al., 2023b; Wang et al., 2023). Although replay-based strategies can effectively preserve old-class performance, their applicability is often limited by privacy constraints, memory budgets, and distribution mismatch between replayed samples and current data.

**Regularization.** Under the *fixed-architecture and no-replay* setting, regularization-based methods have become the dominant paradigm. Knowledge distillation enforces consistency between the current model and a frozen teacher from the previous step—at the logit, feature map, class distribution, or prototype level—effectively mitigating catastrophic forgetting (Michieli & Zanuttigh, 2019; Baek et al., 2022; Zhu et al., 2026; Chen et al., 2025). Beyond distillation, contrastive learning has also been introduced as a regularization mechanism to improve class discriminability. Michieli and Zanuttigh (Michieli & Zanuttigh, 2021) reduced representation overlap between old and new classes via attraction and repulsion forces in the feature space, while Zhao et al. (Zhao et al., 2023) combined contrastive loss with distillation to explicitly enlarge the representation gap between new and similar old classes under an exemplar-free setting.

**Research Gap: Optimization Bottleneck in Low-Margin Regions.** While existing methods effectively stabilize old

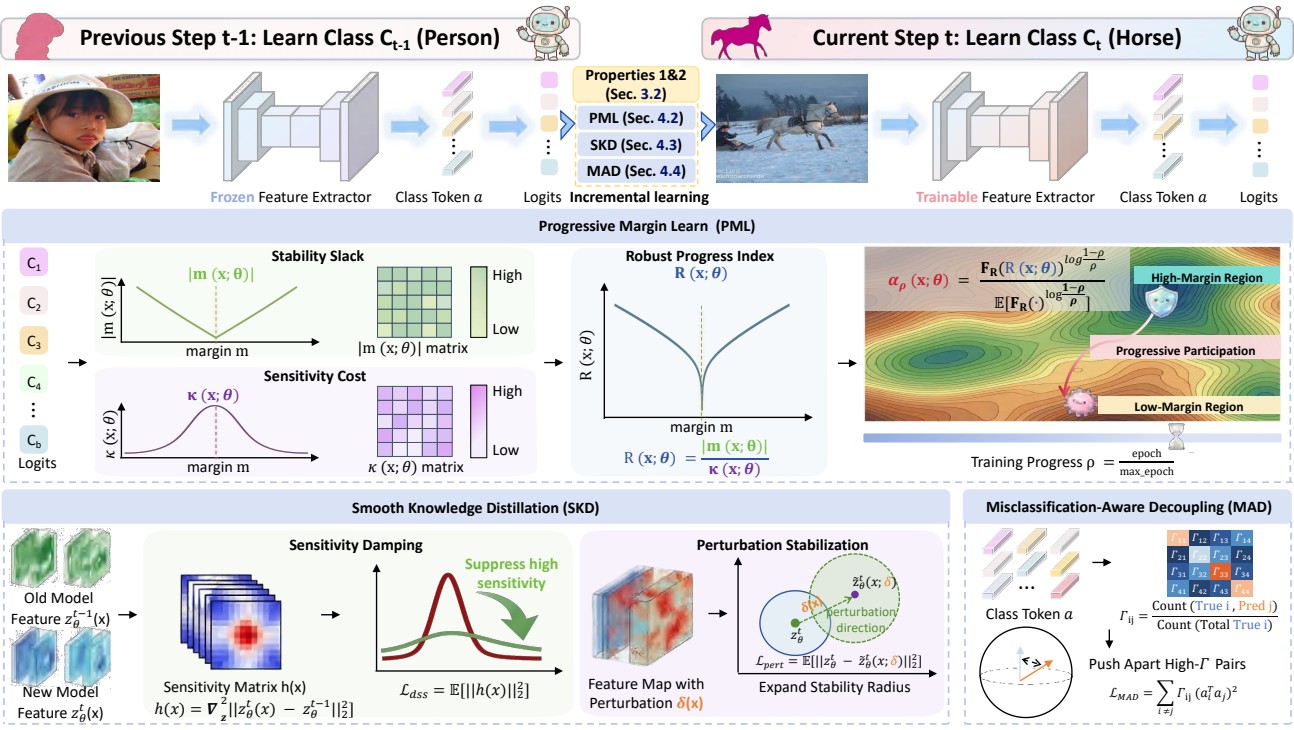

*Figure 3.* **Overview of the Proposed LDKA Method.** At step $t$, guided by the derived low-margin properties (Sec. 3.2), LDKA addresses low-margin optimization via three complementary optimization strategies: PML determines **what to learn** by progressively allocating pixel-level optimization budget, SKD determines **how to learn** by stabilizing low-margin updates, and MAD determines **where to separate** by decoupling highly confused class pairs. Together, they provide an allocation–stabilization–decoupling optimization design.

knowledge, they rarely address the optimization bottleneck in low-margin regions—pixels where the logit of the ground-truth class and the largest competing logit are close, indicating high uncertainty. These pixels hinder new-class expansion and result in confusion between old and new classes. Under a *fixed-architecture and replay-free* setting, we rigorously analyze and justify the underlying reasons why low-margin regions are hard to optimize, and propose a learnability-driven optimization method.

## 3. Theoretical Analysis

### 3.1. Problem Setup

In CISS, the model is trained sequentially over steps $t = 1, \ldots, T$, observing a dataset $\mathcal{D}_t = \{(x_n, y_n^t)\}_{n=1}^{N_t}$, where $x_n$ denotes the $n$-th input image and $y_n^t$ denotes its available annotation at step $t$. Under the incremental setting, $y_n^t$ provides labels only for the newly introduced classes $\mathcal{C}^t$, while pixels belonging to previously learned or future classes are treated as background. The set of previously learned classes is $C^{t-1} = \bigcup_{\tau < t} C^\tau$. The model parameterized by $\theta$ produces pixel-wise logits $s(x;\theta)$ and class probabilities $p(x;\theta) = \text{softmax}(s(x;\theta))$, along with intermediate representations $z_\theta^t(x)$. A frozen model from step $t-1$, denoted by $\theta^{t-1}$, is retained for old-knowledge

preservation. Since ground-truth labels for old classes are unavailable, pseudo-labels $y^*$ (Cermelli et al., 2020) are generated to approximate old-class supervision. This paper identifies low-margin regions as the bottleneck in CISS, which limits the performance of newly learned classes.

### 3.2. Key Properties of Low-Margin Regions

In this paper, we provide an optimization-based diagnosis of low-margin regions in CISS, highlighting their high second-order sensitivity along the margin direction and small stability radius as key factors limiting new-class learning and boundary refinement.

**Low-margin region definition.** For an input $x$ with training label $y^*$, let the most confident competing class be

$$c^*(x) = \arg\max_{j \neq y} s_j(x;\theta), \qquad (1)$$

where $y$ denotes the true class index. The logit margin is defined as:

$$m(x;\theta) = s_y(x;\theta) - s_{c^*(x)}(x;\theta). \qquad (2)$$

Low-margin regions correspond to $m(x;\theta) \approx 0$, where the top-1 and top-2 logits are close, leading to high uncertainty and strong sensitivity to small perturbations.

**Property 1: Maximum Second-Order Sensitivity Along the Margin Direction Occurs in Low-Margin Regions.** When learning new classes at step $t$, we adopt the softmax cross-entropy loss (Mao et al., 2023) as $\mathcal{L}_{\text{seg}}$. Considering the directional second-order sensitivity of $\mathcal{L}_{\text{seg}}$ along the margin direction $m(x, \theta)$, the directional second derivative under softmax is

$$\frac{\partial^2 \mathcal{L}_{\text{seg}}}{\partial m^2} = \left(p_y + p_{c^*}\right) - \left(p_y - p_{c^*}\right)^2. \qquad (3)$$

Here, $p_y$ represents the predicted probability for the true class $y$, and $p_{c^*}$ represents the predicted probability for the most confident competing class $c^*$, where $c^*$ is the class with the second-highest logit. The upper bound of the second derivative is:

$$\frac{\partial^2 \mathcal{L}_{\text{seg}}}{\partial m^2} \lesssim 1, \qquad (4)$$

and the Equation (3) is maximized near $m(x; \theta) \approx 0$. Consequently, low-margin regions exhibit the highest directional second-order sensitivity, such that small incremental updates can cause disproportionate shifts between competing logits, leading to performance degradation in CISS. Derivation details are provided in Appendix A.

**Property 2: Small Stability Radius in Low-Margin Regions.** A small perturbation $\Delta\theta$ changes the margin according to the first-order Taylor approximation (Pötzsche & Rasmussen, 2006):

$$m(x; \theta + \Delta\theta) = m(x; \theta) + \nabla_\theta m(x; \theta)^\top \Delta\theta + \mathcal{O}\left(\|\Delta\theta\|^2\right), \qquad (5)$$

where $\nabla_\theta m(x; \theta)$ is the gradient of the logit margin with respect to the parameters. The stability radius $r$ is the minimum perturbation required to reach the decision boundary:

$$r(x; \theta) \triangleq \inf_{\Delta\theta} \|\Delta\theta\| \quad \text{s.t.} \quad m(x; \theta)\, m(x; \theta + \Delta\theta) \le 0. \qquad (6)$$

Any boundary-reaching perturbation satisfies

$$\|\Delta\theta\| \ge \frac{|m(x; \theta)|}{\|\nabla_\theta m(x; \theta)\|}, \qquad (7)$$

and thus $r(x; \theta) \approx \frac{|m(x;\theta)|}{\|\nabla_\theta m(x;\theta)\|}$. When $m(x; \theta) \approx 0$, the stability radius is small. Consequently, during later incremental steps, learning new classes becomes highly sensitive to small parameter perturbations, which can flip predictions on old classes. Derivation details are provided in Appendix B.

# 4. Method

## 4.1. Overview of the Proposed Method

Motivated by the above properties, we propose Learnability-Driven Knowledge Assimilation (LDKA) with three complementary optimization strategies: as shown in Fig. 3, PML

allocates training focus (what to learn), SKD stabilizes low-margin updates (how to learn), and MAD decouples confused classes (where to separate):

**1. Progressive Margin Learning (PML)**: This strategy guides the training signal to progressively migrate from high-margin regions to low-margin regions. This approach helps the model gradually refine the decision boundary over time.

**2. Smooth Knowledge Distillation (SKD)**: Unlike previous methods that primarily rely on feature difference constraints to mitigate catastrophic forgetting, we introduce high directional second-order sensitivity suppression and local perturbation constraints. This strategy suppresses high-frequency oscillations in the new step's learning process and alleviates unsatisfactory learning in the low-margin region caused by small perturbations.

**3. Misclassification-Aware Decoupling (MAD)**: We introduce a misclassification matrix that quantifies the degree of misclassification between new and old classes. Based on this information, we perform semantic axis decoupling, which promotes the stable contraction of the low-margin region, further improving the learning process.

We optimize the total objective:

$$\mathcal{L}_{\text{total}} = \mathcal{L}_{\text{seg-PML}} + \mathcal{L}_{\text{SKD}} + \mathcal{L}_{\text{MAD}}. \qquad (8)$$

## 4.2. Progressive Margin Learning

**Motivation.** Based on properties 1 and 2, we observe that gradients from low-margin regions exhibit higher sensitivity and lower stability, leading to optimization difficulty during early training. To address this, we progressively engage low-margin pixels throughout training.

**Measure of stability and sensitivity in the margin regions.** To quantify the stability of pixels, we define the *stability slack* as the magnitude of the margin $|m(x; \theta)|$, where a larger margin indicates a more stable boundary. For the *sensitivity cost*, we use the second derivative $\kappa(x; \theta) = \mathcal{L}''_{seg}(m(x; \theta))$, which reaches its peak when $m(x; \theta) \approx 0$. To combine these measures into a unified metric, we define the *Robust Progress Index* $R(x; \theta)$ as:

$$R(x; \theta) = \frac{|m(x; \theta)|}{\kappa(x; \theta)}. \qquad (9)$$

This index balances stability slack and sensitivity, reflecting the pixel's contribution to the optimization process. We then rank all pixels within a mini-batch by their $R(x; \theta)$ values and define the empirical cumulative distribution function (CDF) (Chun et al., 2000) as:

$$u(x) = F_R(R(x; \theta)) \in (0, 1], \qquad (10)$$

where $u(x)$ represents the percentile rank of $R(x; \theta)$ within the batch. Higher values of $u(x)$ correspond to more stable regions, while lower values indicate proximity to low-margin areas with higher sensitivity.

**Measure of the training progress.** To enable progressive participation optimization throughout training, in addition to the above metric for measuring the extent to which pixels belong to high-margin regions, we introduce a monitoring signal based on training progress. Together, these allow the focus to shift from high-margin regions in the early stages to low-margin regions in the later stages of training. We define the raining progress as

$$\rho = \frac{epoch}{max\_epoch}, \qquad (11)$$

and construct a monitoring signal $e(\rho)$ as

$$e(\rho) = \log \frac{1 - \rho}{\rho}. \qquad (12)$$

Since $e(\rho)$ is singular at the endpoints, it is evaluated with $\rho$ bounded within the open interval $(0, 1)$ in practice.

**Supervision with Progressive Participation Weighting.** The pixel participation weight $a_\rho(x)$ is then defined as:

$$a_\rho(x) = \frac{u(x)^{e(\rho)}}{\mathbb{E}[u(\cdot)^{e(\rho)}]}. \qquad (13)$$

When $\rho \to 0$, $e(\rho)$ is large and positive, thus upweighting pixels with larger percentile scores $u(x)$. When $\rho \to 1$, $e(\rho)$ becomes negative, thereby explicitly upweighting smaller $u(x)$ (i.e., low-margin regions) for result refinement. We apply the progressive pixel participation weights directly to the pixel-wise cross-entropy loss:

$$\mathcal{L}_{\text{seg-PML}} = \mathbb{E}_{(x,y) \sim D_t} \left[ a_\rho(x) \cdot \mathcal{L}_{\text{CE}}(p_\theta(\cdot \mid x), y^*) \right]. \quad (14)$$

This optimization strategy reallocates the optimization budget over time, initially prioritizing stable pixels and gradually incorporating low-margin pixels as training progresses.

### 4.3. Smooth Knowledge Distillation

**Motivation.** Existing distillation methods primarily constrain high-confidence regions, leaving low-margin pixels insufficiently regulated. As shown in Property 1 and 2, these pixels exhibit high directional second-order sensitivity and a small stability radius, making them highly unstable under incremental updates. We therefore extend distillation to suppress directional second-order sensitivity and improve robustness in low-margin regions via structured representation-level perturbations.

**Standard Knowledge Distillation.** The standard KD loss for CISS (Hinton et al., 2015; Rebuffi et al., 2017) is:

$$\mathcal{L}_{\text{KD}} = \mathbb{E}_x \left[ \|z^t(x) - z^{t-1}(x)\|_2^2 \right], \qquad (15)$$

where $z^t(x)$ and $z^{t-1}(x)$ are the feature representations at timestep $t$ and $t - 1$, respectively. This loss helps retain knowledge from previous stages by minimizing the feature difference between the current and previous model.

**Directional second-order sensitivity damping.** To mitigate high-frequency oscillations, we impose a directional second-order sensitivity penalty on top of the standard knowledge distillation:

$$h(x) = \nabla_{z_\theta^\mathcal{L}}^2 \left( \|z^t(x) - z^{t-1}(x)\|_2^2 \right), \qquad (16)$$

with the associated loss:

$$\mathcal{L}_{\text{dss}} = \mathbb{E}_x \left[ \|h(x)\|_2^2 \right]. \qquad (17)$$

This loss explicitly suppresses high directional second-order sensitivity, reducing high-frequency oscillations in the model's updates and allowing for more consistent accumulation of knowledge over time.

**Perturbation stabilization.** Low-margin regions have a small stability radius and are therefore sensitive to local feature perturbations during incremental updates. To improve the stability of these regions, we introduce a feature-space perturbation regularizer. Specifically, we denote the intermediate representation $z^t(x)$ as the propagated response of a pre-decoder feature $f^t(x)$ through the subsequent decoder mapping $H_\phi(\cdot)$. The propagated response of the current model is thus written as

$$z^t(x) = H_\phi(f^t(x)). \qquad (18)$$

The perturbation direction is constructed from the gradient of the distillation discrepancy with respect to the intermediate feature representation:

$$d(x) = \nabla_{f^t(x)} \left\| z^t(x) - z^{t-1}(x) \right\|_2^2, \qquad (19)$$

where $z^{t-1}(x)$ denotes the frozen response from the previous step. We normalize this direction to control the perturbation magnitude:

$$\delta^t(x) = \frac{d(x)}{\|d(x)\|_2}. \qquad (20)$$

The perturbation is injected into the intermediate feature representation and then propagated through the decoder:

$$\tilde{z}^t(x) = H_\phi\big(f^t(x) + \delta^t(x)\big). \qquad (21)$$

We then constrain the propagated responses before and after perturbation to be consistent:

$$\mathcal{L}_{\text{pert}} = \mathbb{E}_x \left[ \left\| H_\phi(f^t(x)) - H_\phi\big(f^t(x) + \delta^t(x)\big) \right\|_2^2 \right]. \qquad (22)$$

*Table 1.* Comparative experiments on VOC dataset. **Bold** and underlined values denote the best and second-best results, respectively. The † symbol indicates results reproduced using the same version of ViT. − indicates that the relevant experiment is not mentioned in the original papers. Across five incremental settings, our method achieves the highest overall mIoU, demonstrating consistent superiority.

| Method | Backbone | 15-5 (2 steps) | | | 19-1 (2 steps) | | | 15-1 (6 steps) | | | 2-2 (10 steps) | | | 10-1 (11 steps) | | |
|---|---|---|---|---|---|---|---|---|---|---|---|---|---|---|---|---|
| | | 0-15 | 16-20 | All | 0-19 | 20 | All | 0-15 | 16-20 | All | 0-2 | 3-20 | All | 0-10 | 11-20 | All |
| MIB (Cermelli et al., 2020) | ResNet101 | 76.4 | 50.0 | 70.1 | 71.4 | 23.6 | 69.1 | 34.2 | 13.5 | 29.3 | 41.1 | 23.4 | 25.9 | 12.3 | 13.1 | 12.7 |
| PLOP (Douillard et al., 2021) | ResNet101 | 75.7 | 51.7 | 70.0 | 75.4 | 37.4 | 73.6 | 65.1 | 21.1 | 54.6 | 24.1 | 11.9 | 13.6 | 44.0 | 15.5 | 30.4 |
| SSUL (Cha et al., 2021) | ResNet101 | 78.4 | 55.8 | 73.0 | 77.8 | 49.8 | 76.5 | 78.4 | 49.0 | 71.4 | – | – | – | 74.0 | 53.2 | 64.1 |
| MicroSeg (Zhang et al., 2022b) | ResNet101 | 82.0 | 59.2 | 76.6 | 79.3 | 62.9 | 78.5 | 81.3 | 52.5 | 74.4 | 60.0 | 50.9 | 52.2 | 77.2 | 57.2 | 67.7 |
| RCIL (Zhang et al., 2022a) | ResNet101 | 78.8 | 52.0 | 72.4 | 68.5 | 12.1 | 65.8 | 70.6 | 23.7 | 59.4 | 28.3 | 19.0 | 20.3 | 55.4 | 15.1 | 36.2 |
| LGKD (Yang et al., 2023) | ResNet101 | 79.5 | 54.8 | 73.6 | 77.3 | 42.9 | 75.7 | 70.6 | 30.9 | 61.1 | – | – | – | – | – | – |
| CoMasTRe (Gong et al., 2024) | ResNet101 | 79.7 | 51.9 | 73.1 | 75.1 | 69.5 | 74.8 | 69.8 | 43.6 | 63.6 | – | – | – | – | – | – |
| Adapter (Zhu et al., 2025) | ResNet101 | – | – | – | – | – | – | 79.9 | 51.9 | 73.2 | 62.8 | 57.9 | 58.6 | 74.9 | 54.3 | 65.1 |
| MIB† (Cermelli et al., 2020) | ViT | 78.5 | 63.2 | 74.9 | 80.4 | 47.8 | 78.8 | 72.6 | 23.5 | 60.9 | 41.1 | 29.3 | 31.0 | 11.4 | 18.9 | 15.0 |
| SSUL† (Cha et al., 2021) | ViT | 79.7 | 55.3 | 73.9 | 80.8 | 31.5 | 78.5 | 78.1 | 33.4 | 67.5 | 60.3 | 40.6 | 43.4 | 74.3 | 51.0 | 63.2 |
| MicroSeg† (Zhang et al., 2022b) | ViT | 81.9 | 54.0 | 75.3 | 79.0 | 25.3 | 76.4 | 80.5 | 40.8 | 71.0 | 64.8 | 43.4 | 46.5 | 73.5 | 53.0 | 63.7 |
| Incrementer (Shang et al., 2023) | ViT | – | – | – | – | – | – | 79.6 | 59.6 | 74.8 | – | – | – | 77.6 | 60.3 | 69.4 |
| CoinSeg (Zhang et al., 2023) | ViT | 82.1 | 63.2 | 77.6 | 81.5 | 44.8 | 79.8 | 82.7 | 52.5 | 75.5 | 70.1 | 63.3 | 64.3 | 80.1 | 60.0 | 70.5 |
| MBS† (Park et al., 2024) | ViT | 83.9 | 72.6 | 81.2 | 82.2 | **72.6** | 81.8 | 81.9 | 65.6 | 78.0 | 67.5 | 73.4 | 72.6 | 80.0 | 72.9 | 76.6 |
| Nest (Xie et al., 2024) | ViT | 81.2 | 67.4 | 77.9 | 79.7 | 60.0 | 78.8 | 77.0 | 53.3 | 71.4 | – | – | – | 65.2 | 35.8 | 51.2 |
| CoGaMiD (Zhu et al., 2026) | ViT | – | – | – | – | – | – | **83.2** | 61.2 | 77.8 | **73.4** | 70.0 | 70.5 | **81.1** | 65.9 | 73.9 |
| **LDKA (Ours)** | **ViT** | **84.6** | **75.3** | **82.4** | **82.9** | 73.9 | **82.5** | 83.3 | **69.9** | **80.1** | 72.3 | **75.8** | **75.3** | 80.0 | **73.1** | **76.7** |
| Joint (Upper bound) | ViT | 85.5 | 80.3 | 84.3 | 84.4 | 79.6 | 84.2 | 83.9 | 79.1 | 82.8 | 77.3 | 85.5 | 84.3 | 85.0 | 84.7 | 84.9 |

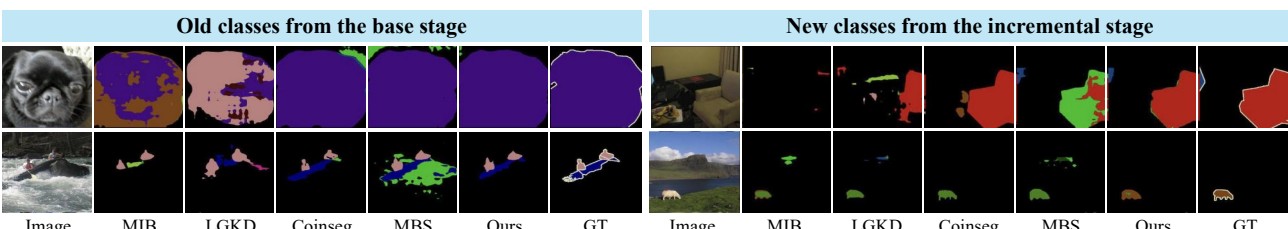

*Figure 4.* Qualitative results under the 15-1 setting. Results from the base stage and the incremental stage show that our method achieves more accurate pixel-level segmentation of old classes with resistance to forgetting, while reducing misclassification of new-class pixels.

This regularizer suppresses unstable response variations induced by local feature perturbations, thereby improving the robustness during incremental learning.

**Overall SKD Loss.** The total SKD loss is:

$$\mathcal{L}_{\text{SKD}} = \mathcal{L}_{\text{dss}} + \mathcal{L}_{\text{pert}}. \qquad (23)$$

### 4.4. Misclassification-Aware Decoupling

**Motivation.** In class-incremental learning, new classes often share significant similarity with old classes, leading to potential misclassification in low-margin regions. To address this issue, we propose a Misclassification-Aware Decoupling approach, which explicitly quantifies and reduces the overlap between class representations by considering misclassification effects.

**Measure of class conflict.** To measure the degree of misclassification ratio between class $i$ and class $j$, the competition matrix $\Gamma_{ij}$ is calculated per epoch as follows:

$$\Gamma_{ij} = \frac{\text{Count}(\text{True class } i, \text{Predicted as class } j)}{\text{Count}(\text{Total true class } i)} \qquad (24)$$

**Overall MAD Loss.** Based on the updated competition matrix, we define the decoupling loss $\mathcal{L}_{MAD}$ as:

$$\mathcal{L}_{\text{MAD}} = \sum_{i \neq j} \Gamma_{ij} \left( a_i^\top a_j \right)^2, \qquad (25)$$

where $a_i$ and $a_j$ are the class tokens (Dosovitskiy et al., 2021) of classes $i$ and $j$, respectively. By specifically accounting for the misclassification effect, the model increases the distance between class tokens that are likely to be misclassified. This helps ensure that the learning of new classes does not interfere with the representation of old classes, leading to more reliable class representations in CISS.

## 5. Experiment

### 5.1. Experimental Setup

**Datasets, evaluation metric, and protocols.** Following prior works on class-incremental semantic segmentation (CISS), we evaluate our method on Pascal VOC 2012 (Everingham et al., 2010) (20 classes) and ADE20K (Zhou et al., 2017) (150 classes), using Mean Intersection over Union (mIoU) as the evaluation metric. The incremental

*Table 2.* Comparative experiments on ADE20K. Our method is capable of effectively learning new knowledge and resisting catastrophic forgetting without accessing old-class data for rehearsal. Notably, the overall performance of our method across the four incremental settings is very close to that of joint training, which is commonly regarded as the upper bound of performance in CISS.

| Method | Backbone | 100–50 (2 steps) | | | 50–50 (3 steps) | | | 100–10 (6 steps) | | | 100–5 (11 steps) | | |
|---|---|---|---|---|---|---|---|---|---|---|---|---|---|
| | | 0-100 | 101-150 | All | 0-50 | 51-150 | All | 0-100 | 101-150 | All | 0-100 | 101-150 | All |
| SDR (Michieli & Zanuttigh, 2021) | ResNet101 | 37.5 | 25.5 | 33.5 | 42.9 | 25.4 | 31.3 | 28.9 | 11.7 | 23.2 | 36.7 | 5.7 | 26.4 |
| PLOP (Douillard et al., 2021) | ResNet101 | 41.9 | 14.9 | 33.0 | 48.8 | 21.0 | 30.4 | 40.5 | 13.6 | 31.6 | 39.1 | 7.8 | 28.7 |
| SSUL (Cha et al., 2021) | ResNet101 | 41.3 | 18.0 | 33.6 | 48.4 | 20.2 | 29.7 | 40.2 | 18.8 | 33.1 | 39.9 | 17.4 | 32.4 |
| REMINDER (Phan et al., 2022) | ResNet101 | 41.6 | 19.2 | 34.2 | 47.1 | 20.4 | 29.4 | 39.0 | 21.3 | 33.1 | 36.1 | 16.4 | 29.6 |
| Microseg (Zhang et al., 2022b) | ResNet101 | 40.2 | 18.8 | 33.1 | 48.6 | 24.8 | 32.8 | 41.5 | 21.6 | 34.9 | 40.4 | 20.5 | 33.8 |
| IDEC (Zhao et al., 2023) | ResNet101 | 42.0 | 18.2 | 34.1 | 47.4 | 26.0 | 33.2 | 42.3 | 17.6 | 34.1 | 39.2 | 14.6 | 31.1 |
| LGKD (Yang et al., 2023) | ResNet101 | 43.4 | 25.7 | 37.5 | 48.9 | 29.4 | 36.0 | 41.9 | 22.0 | 35.3 | - | - | - |
| LAG (Yuan et al., 2024) | ResNet101 | 41.6 | 19.7 | 34.3 | 47.7 | 26.1 | 33.4 | 41.0 | 18.7 | 33.6 | 40.0 | 17.2 | 32.5 |
| CoMasTRe (Gong et al., 2024) | ResNet101 | 45.7 | 26.0 | 39.2 | - | - | - | 42.3 | 18.4 | 34.4 | 40.8 | 15.8 | 32.5 |
| Adapter (Zhu et al., 2025) | ResNet101 | 43.1 | 23.6 | 36.6 | 49.3 | 27.3 | 34.7 | 42.9 | 19.9 | 35.3 | 42.6 | 18.0 | 34.5 |
| MIB† (Cermelli et al., 2020) | ViT | 46.4 | 35.0 | 42.6 | 52.2 | 35.6 | 41.2 | 43.0 | 30.8 | 39.0 | 40.2 | 26.6 | 35.7 |
| SSUL† (Cha et al., 2021) | ViT | 41.9 | 20.1 | 34.7 | 49.5 | 21.3 | 30.8 | 40.7 | 19.0 | 33.5 | 41.3 | 16.0 | 32.9 |
| Microseg† (Zhang et al., 2022b) | ViT | 41.1 | 24.1 | 35.5 | 49.8 | 23.9 | 32.6 | 41.0 | 22.6 | 34.9 | 41.2 | 21.0 | 34.5 |
| Coinseg (Zhang et al., 2023) | ViT | 41.6 | 26.7 | 36.7 | 49.0 | 28.9 | 35.7 | 42.1 | 24.5 | 36.3 | 43.1 | 24.1 | 36.8 |
| CoMFormer (Cermelli et al., 2023) | ViT | 44.7 | 26.2 | 38.6 | - | - | - | 40.6 | 15.6 | 32.3 | 39.5 | 13.6 | 30.9 |
| MBS† (Park et al., 2024) | ViT | **49.4** | _37.5_ | _45.5_ | _55.6_ | _39.8_ | _45.1_ | 48.1 | _35.2_ | _43.8_ | _45.7_ | 22.7 | _38.1_ |
| Nest (Xie et al., 2024) | ViT | 42.8 | 27.8 | 37.8 | 49.7 | 29.3 | 36.2 | 41.8 | 23.8 | 35.8 | 40.5 | 19.9 | 33.7 |
| CoGaMiD (Zhu et al., 2026) | ViT | 43.9 | 27.3 | 38.4 | 49.9 | 29.8 | 36.6 | **49.9** | 26.5 | 42.2 | 43.6 | _25.8_ | 37.7 |
| LDKA (Ours) | ViT | _49.1_ | **40.7** | **46.3** | **55.8** | **42.3** | **46.9** | 48.0 | **38.5** | **44.9** | 46.1 | **28.1** | **40.1** |
| Joint (Upper bound) | ViT | 52.9 | 42.6 | 49.5 | 58.9 | 44.7 | 49.5 | 52.7 | 42.4 | 49.3 | 52.6 | 42.6 | 49.3 |

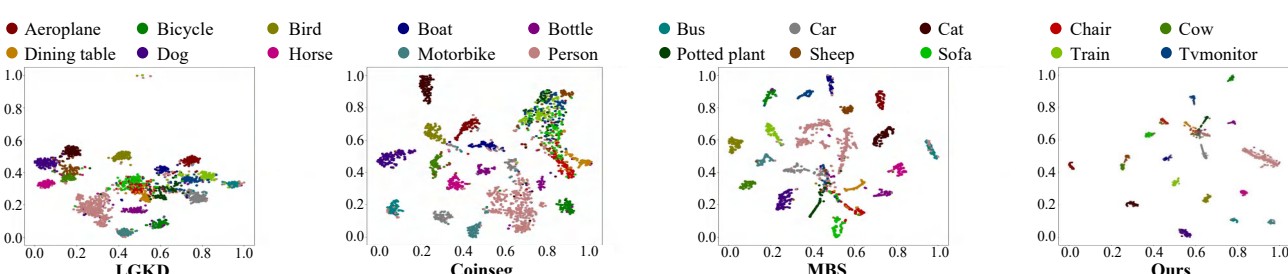

*Figure 5.* Qualitative analysis under the 15-1 configuration using the T-SNE plot. Our method shows better intra-class clustering than recent approaches, suggesting greater potential for accommodating new classes in the future, which is crucial for CISS.

learning scenario is defined as $N_b - N_n$, where $N_b$ is the base class count and $N_n$ is the number of new classes per step. For example, in the 15–1 setting, the model learns 15 base classes and one new class per step, totaling six steps for a 20-class dataset. We focus on the overlapped setting, where the background includes both past and future classes, as it better reflects practical CISS.

**Implementation details.** We use a ViT-B/16 model (Dosovitskiy et al., 2021) pretrained on ImageNet (Deng et al., 2009) as the encoder and a two-layer transformer decoder with $512 \times 512$ resolution. The model is optimized with SGD (Ketkar, 2017) (momentum 0.9, weight decay $1 \times 10^{-5}$) for 64 epochs. The initial learning rate is $1 \times 10^{-3}$ for the base stage, adjusted to $1 \times 10^{-4}$ for Pascal VOC and $5 \times 10^{-4}$ for ADE20K during incremental learning. All experiments are conducted in PyTorch on a workstation with four NVIDIA RTX 3090 GPUs.

## 5.2. Comparison with State-of-the-Art Methods

**Quantitative analysis on Pascal VOC and ADE20k.** We evaluate our method on Pascal VOC 2012 under five incremental configurations, as shown in Tab. 1, and compare it with recent state-of-the-art CISS methods. "Joint" represents the oracle setting where all classes are trained simultaneously. In the 2-2 scenario, the new class performance improves by 2.4, and in the 15-1 configuration, it improves by 4.3. On the challenging ADE20K dataset, in short-term settings (100–50 and 50–50), new class mIoU improves by 3.2 and 2.5, respectively, while reducing forgetting of old classes, as shown in Tab. 2. In long-term settings (100–10 and 100–5), the improvements of new classes are 3.3 and 2.3. These results demonstrate that our method effectively mitigates catastrophic forgetting and enhances plasticity in both short-term and long-term incremental scenarios.

**Qualitative Analysis on Results Comparison and T-SNE.** In Fig. 4, we visualize the segmentation results of our

*Table 3.* Ablation study of components on Pascal VOC 15-1. Integrating all components yields the best performance.

| Num | Baseline | $\mathcal{L}_{\text{seg-PML}}$ | $\mathcal{L}_{\text{SKD}}$ | $\mathcal{L}_{\text{MAD}}$ | 15-1 (6 steps) | | |
|---|---|---|---|---|---|---|---|
| | | | | | 0–15 | 16–20 | All |
| 1 | ✓ | | | | 71.8 | 45.0 | 65.4 |
| 2 | ✓ | ✓ | | | 82.2 | 61.4 | 77.2 |
| 3 | ✓ | ✓ | ✓ | | 83.0 | 68.4 | 79.5 |
| 4 | ✓ | ✓ | ✓ | ✓ | 83.3 | 69.9 | 80.1 |

*Table 4.* Ablation study about the design of $\mathcal{L}_{\text{MAD}}$. $\mathcal{L}_{\text{MAD}}$ is more suitable for CISS than recent inter-class decoupling methods.

| | 15-1 (6 steps) | | |
|---|---|---|---|
| | 0–15 | 16–20 | All |
| Contrastive learning | 82.0 | 67.9 | 78.5 |
| Contrastive learning with $\Gamma_{ij}$ | 82.2 | 68.6 | 78.8 |
| Orthogonal | 82.0 | 63.4 | 77.4 |
| LDKA (Ours) | 82.7 | 69.9 | 79.5 |

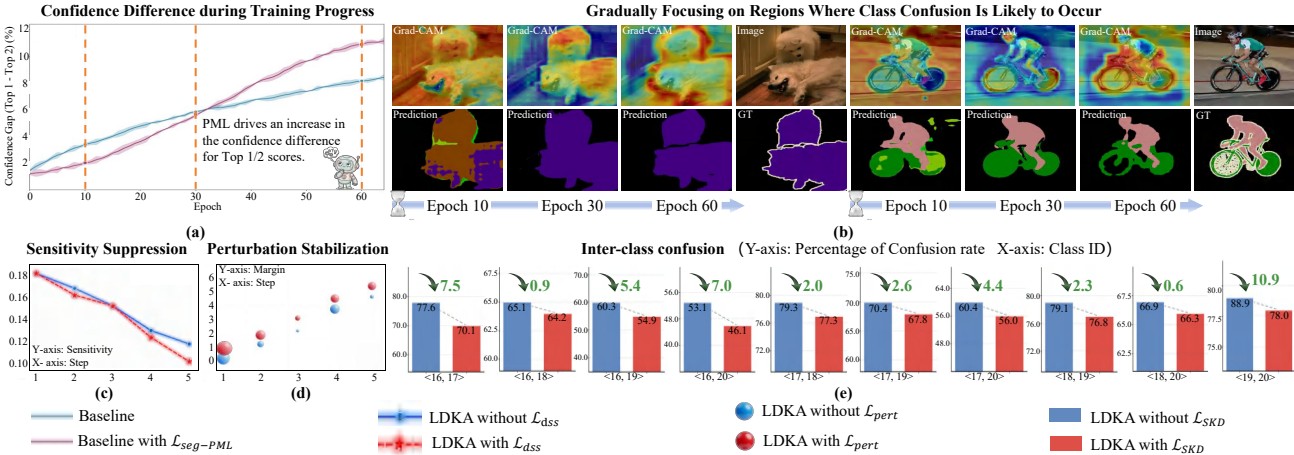

*Figure 6.* (a)–(b) analyze the effect of $\mathcal{L}_{\text{seg-PML}}$; (c)–(e) analyze $\mathcal{L}_{\text{dss}}$, $\mathcal{L}_{\text{pert}}$, and $\mathcal{L}_{\text{SKD}}$, respectively. $\mathcal{L}_{\text{seg-PML}}$ enlarges the top-1/top-2 confidence gap, while $\mathcal{L}_{\text{dss}}$ and $\mathcal{L}_{\text{pert}}$ suppress directional second-order sensitivity and increase the stability radius. $\mathcal{L}_{\text{SKD}}$ further mitigates class confusion.

method and recent state-of-the-art approaches under the 15-1 configuration. The left side of the figure shows the segmentation of old classes (0-15) during the base stage, while the right side shows the segmentation results of newly learned classes (16-20) during the incremental stage. Our method achieves superior pixel-level segmentation without relying on replay or dynamic architecture expansion. Furthermore, the T-SNE results in Fig. 5 reveal that our method exhibits a sparser inter-class distribution and tighter intra-class clustering. This indicates that our approach leaves more space for learning future new classes, which is crucial for class-incremental semantic segmentation.

### 5.3. Ablation Studies

**Effectiveness of Component Analysis in LDKA.** Tab. 3 presents an ablation study on Pascal VOC 2012, evaluating the impact of the three core components: Progressive Margin Learning ($\mathcal{L}_{\text{seg-PML}}$), Smooth Knowledge Distillation ($\mathcal{L}_{\text{SKD}}$), and Misclassification-Aware Decoupling ($\mathcal{L}_{\text{MAD}}$). The baseline (Num. 1) uses pseudo-label–guided cross-entropy loss (Cermelli et al., 2020) to address background drift. Num. 2 adds $\mathcal{L}_{\text{seg-PML}}$, improving stability and plasticity by guiding learning from high- to low-margin regions. In Num. 3, $\mathcal{L}_{\text{SKD}}$ further constrains high second-order sensitivity along the margin direction, enhancing robustness

in low-margin areas. Num. 4 includes $\mathcal{L}_{\text{MAD}}$, reducing inter-class confusion and improving performance on new classes. The model incorporating all components achieves the best performance, highlighting the complementary roles of $\mathcal{L}_{\text{seg-PML}}$, $\mathcal{L}_{\text{SKD}}$, and $\mathcal{L}_{\text{MAD}}$.

**Impact of Progressive Margin Learning.** To analyze the effect of Progressive Margin Learning ($\mathcal{L}_{\text{seg-PML}}$), we measure the confidence gap between the top-1 and top-2 predicted classes, reflecting the separation of competing logits. Fig. 6(a) shows the evolution of this metric across epochs for the baseline model ( ▬ ) and the model with $\mathcal{L}_{\text{seg-PML}}$ ( ▬ ). The results show that $\mathcal{L}_{\text{seg-PML}}$ progressively enlarges the Top1–Top2 confidence gap, strengthening decision margins. Grad-CAM visualizations in Fig. 6(b) further illustrate the model gradually reallocates attention to margin-critical areas over time, resulting in more refined predictions.

**Impact of Smooth Knowledge Distillation.** Fig. 6(c) shows the evolution of second-order sensitivity along the margin direction during training with ( --✱-- ) and without ( ▬●▬ ) the proposed second-order sensitivity damping along the margin direction $\mathcal{L}_{\text{dss}}$. Incorporating $\mathcal{L}_{\text{dss}}$ reduces second-order sensitivity along the margin direction, indicating effective directional second-order sensitivity sup-

pression. To assess the effectiveness of $\mathcal{L}_{\text{pert}}$ in enhancing stability, we visualize bubble plots with ( 🔴 ) and without ( 🔵 ) $\mathcal{L}_{\text{pert}}$ in Fig. 6(d). A larger bubble plot indicates a larger stability radius. The results show that $\mathcal{L}_{\text{pert}}$ increases the stability radius and maintains a larger logit margin at each incremental step. In Fig. 6(e), we compare the performance of $\mathcal{L}_{\text{SKD}}$ (combined with $\mathcal{L}_{\text{dss}}$ and $\mathcal{L}_{\text{pert}}$) with ( 🟥 ) and without ( 🟦 ) $\mathcal{L}_{\text{SKD}}$. The inter-class confusion rates among classes 16–20 are consistently reduced, with decreases of 7.5, 0.9, 5.4, 7.0, 2.0, 2.6, 4.4, 2.3, 0.6, and 10.9 percentage points. The results show that $\mathcal{L}_{\text{SKD}}$ effectively reduces inter-class confusion.

**Ablation about the design of $\mathcal{L}_{\text{MAD}}$.** To evaluate the proposed decoupling strategy, we conduct an ablation study with different class decoupling strategies: (i) standard contrastive learning (Zhang et al., 2023), (ii) contrastive learning enhanced with our competition matrix $\Gamma_{ij}$, and (iii) orthogonality constraints (Liu et al., 2023). As shown in Tab. 4, Incorporating $\Gamma_{ij}$ into contrastive learning strengthens the penalties on frequently misclassified class pairs, improving mIoU by 0.2 for base classes and 0.7 for incremental classes. The orthogonality-based strategy suppresses semantically related features, leading to poorer performance. Overall, $\mathcal{L}_{\text{MAD}}$ outperforms all variants, surpassing the orthogonality-based strategy with mIoU gains of 0.7 on old classes and 6.5 on new classes, effectively reducing class interference.

## 6. Conclusion

This paper addresses the optimization challenges in class-incremental semantic segmentation, focusing on low-margin regions. These regions, characterized by high second-order sensitivity along the margin direction and small stability radius, present a significant bottleneck, hindering the effective expansion and refinement of new-class decision regions. To tackle this, we propose *Learnability-Driven Knowledge Assimilation (LDKA)*, an optimization method targeting low-margin optimization through three key components: Progressive Margin Learning (PML) progressively shifts the optimization focus from high-margin to low-margin regions; Smooth Knowledge Distillation (SKD) suppresses instability caused by second-order sensitivity along the margin direction and enhances robustness; and Misclassification-Aware Decoupling (MAD) reduces interference among classes. Comprehensive experiments and analysis on standard CISS benchmarks demonstrate that LDKA consistently improves stability and plasticity across nine different incremental configurations.

## Acknowledgements

This work was supported in part by the National Natural Science Foundation of China under Grant U24B20139 and in part by the Hubei Provincial Natural Science Foundation of China under Grant 2026AFA040. The computation was completed on the HPC Platform at Huazhong University of Science and Technology.

## Impact Statement

This paper aims to advance machine learning research on class-incremental semantic segmentation, with the goal of learning new semantic classes without direct access to previously seen data. Such a capability can benefit practical applications where storing or revisiting historical data is constrained by privacy, security, or resource limitations, including scenarios in autonomous systems, medical imaging, and remote sensing. Imperfect adaptation during incremental updates can still cause mis-segmentation of both previously learned and newly introduced classes, potentially leading to adverse outcomes in safety-critical domains. To mitigate these concerns, our work emphasizes stable optimization and balanced old–new learning, and we recommend careful validation under diverse incremental settings. We are not aware of direct negative societal impacts specific to the proposed method. However, as with machine learning models in general, if the method is deployed in high-stakes scenarios such as autonomous driving, medical imaging, or remote sensing monitoring, careful validation remains necessary to reduce the risk of adverse outcomes caused by model errors.

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

## Appendix A: Detailed Derivation of Property 1 (Maximum Margin-Directional Second-Order Sensitivity Occurs in Low-Margin Regions.)

**Setting and notation.** For an input sample $x$, $s(x; \theta) \in \mathbb{R}^C$ denotes the logit vector and

$$p_j(x; \theta) \triangleq \hat{y}_\theta(y = j \mid x) = \frac{\exp(s_j(x; \theta))}{\sum_{k=1}^{C} \exp(s_k(x; \theta))}. \tag{26}$$

The ground-truth class index is denoted by $y$ (to avoid symbol conflict, we use $y$ for the class index and $p_j$ for probabilities). The most confident competing class is defined as

$$c^*(x) = \arg\max_{j \neq y} s_j(x; \theta). \tag{27}$$

Following the main paper, the logit margin is defined as

$$m(x; \theta) = s_y(x; \theta) - s_{c^*(x)}(x; \theta). \tag{28}$$

### A.1 Gradient of Cross-Entropy with Respect to Logits

$$\mathcal{L}_{\text{seg}}(x, y; \theta) = -\sum_{j=1}^{C} \mathbb{I}[j = y] \log p_j(x; \theta), \tag{29}$$

where $\mathbb{I}[\cdot]$ denotes the indicator function, i.e., $\mathbb{I}[j = y] = 1$ if $j = y$ and $0$ otherwise. Using standard softmax identities, its gradient with respect to (w.r.t.) the logit $s_j$ is

$$\frac{\partial \mathcal{L}_{\text{seg}}}{\partial s_j} = p_j - \mathbb{I}[j = y]. \tag{30}$$

### A.2 Hessian with Respect to Logits

Differentiate $p_j$ w.r.t. $s_k$:

$$\frac{\partial p_j}{\partial s_k} = p_j(\delta_{jk} - p_k), \tag{31}$$

where $\delta_{jk}$ is the Kronecker delta. Since $\partial \mathcal{L}_{\text{seg}}/\partial s_j = p_j - \mathbb{I}[j = y]$ and $\mathbb{I}[j = y]$ is constant, the Hessian entries are

$$\frac{\partial^2 \mathcal{L}_{\text{seg}}}{\partial s_j \, \partial s_k} = \frac{\partial p_j}{\partial s_k} = p_j(\delta_{jk} - p_k). \tag{32}$$

Equivalently, in matrix form,

$$\nabla_s^2 \mathcal{L}_{\text{seg}} = \text{diag}(p) - pp^\top. \tag{33}$$

### A.3 Directional Second-Order Sensitivity Along the Margin Direction $m = s_y - s_{c^*}$

To study the sensitivity around the decision boundary, we consider the second derivative of $\mathcal{L}_{\text{seg}}$ along the *margin direction* $m = s_y - s_{c^*}$. Define the direction vector $v \in \mathbb{R}^C$ as

$$v_j \triangleq \frac{\partial m}{\partial s_j} = \begin{cases} 1, & j = y, \\ -1, & j = c^*, \\ 0, & \text{otherwise.} \end{cases} \tag{34}$$

The second derivative with respect to $m$ characterizes the margin-directional second-order sensitivity.

$$\frac{\partial^2 \mathcal{L}_{\text{seg}}}{\partial m^2} = v^\top \left( \nabla_s^2 \mathcal{L}_{\text{seg}} \right) v = v^\top \left( \text{diag}(p) - pp^\top \right) v. \tag{35}$$

We expand the two terms separately.

**(i) The diagonal term.**

$$v^\top \mathrm{diag}(p)v = \sum_{j=1}^{C} p_j v_j^2 = p_y + p_{c^*}, \tag{36}$$

because only $v_y^2 = v_{c^*}^2 = 1$ are nonzero.

**(ii) The rank-one term.**

$$v^\top p p^\top v = (p^\top v)^2 = (p_y - p_{c^*})^2. \tag{37}$$

Combining (i) and (ii), we obtain an *exact* expression under the softmax model:

$$\boxed{\frac{\partial^2 \mathcal{L}_{\text{seg}}}{\partial m^2} = (p_y + p_{c^*}) - (p_y - p_{c^*})^2.} \tag{38}$$

### A.4 Upper Bound and Maximizer: Why the Maximum Occurs Near $m \approx 0$

Define $S \triangleq p_y + p_{c^*} \in (0, 1]$ and normalize the top-2 probability mass as

$$\tilde{p}_y \triangleq \frac{p_y}{S}, \qquad \tilde{p}_{c^*} \triangleq \frac{p_{c^*}}{S} = 1 - \tilde{p}_y. \tag{39}$$

Then $p_y = S\tilde{p}_y$ and $p_{c^*} = S(1 - \tilde{p}_y)$, and (38) becomes

$$\frac{\partial^2 \mathcal{L}_{\text{seg}}}{\partial m^2} = S - S^2(2\tilde{p}_y - 1)^2 = S(1 - S) + 4S^2 \, \tilde{p}_y(1 - \tilde{p}_y). \tag{40}$$

**Top-2 dominant regime (low-margin region).** In low-margin regions, the decision is dominated by the top-1 and top-2 classes and the remaining classes contribute negligible probability mass, i.e.,

$$S = p_y + p_{c^*} \approx 1. \tag{41}$$

Under this regime,

$$\frac{\partial^2 \mathcal{L}_{\text{seg}}}{\partial m^2} \approx 4 \, \tilde{p}_y(1 - \tilde{p}_y). \tag{42}$$

The quadratic function $\tilde{p}_y(1 - \tilde{p}_y)$ is maximized at $\tilde{p}_y = \frac{1}{2}$, yielding

$$\tilde{p}_y(1 - \tilde{p}_y) \leq \frac{1}{4}. \tag{43}$$

Substituting into (42) gives

$$\boxed{\frac{\partial^2 \mathcal{L}_{\text{seg}}}{\partial m^2} \lesssim 1, \quad \text{and it is maximized when } \tilde{p}_y \approx \tilde{p}_{c^*} \approx \frac{1}{2}.} \tag{44}$$

**Connection to $m \approx 0$.** When the top-1 and top-2 logits are close (i.e., $m = s_y - s_{c^*} \approx 0$), their softmax probabilities are also close, implying $p_y \approx p_{c^*}$ and hence $\tilde{p}_y \approx \frac{1}{2}$. Therefore, the second-order sensitivity along the margin direction tends to be highest near the decision boundary, corresponding to the low-margin region.

### A.5 Takeaway for CISS

The above derivation shows that, under the standard softmax cross-entropy loss used in our model, the margin-directional second-order sensitivity becomes larger as the competition between the top-1 and top-2 classes intensifies, and is typically highest near $m \approx 0$. This provides an optimization-based explanation for why low-margin pixels are more sensitive to parameter updates and thus constitute a critical bottleneck in class-incremental learning.

### A.6 Connection to Representation-Level Stabilization

The above derivation characterizes the directional second-order sensitivity of $\mathcal{L}_{\text{seg}}$ along the logit-margin direction. To make its connection to the representation-level stabilizer explicit, we view the margin $m$ as a function of the intermediate representation $f$. By the Hessian chain rule, the Hessian of the segmentation loss with respect to $f$ can be written as

$$\nabla_f^2 \mathcal{L}_{\text{seg}} = \frac{\partial^2 \mathcal{L}_{\text{seg}}}{\partial m^2} (\nabla_f m)(\nabla_f m)^\top + \frac{\partial \mathcal{L}_{\text{seg}}}{\partial m} \nabla_f^2 m. \tag{45}$$

This relation shows that the margin-space second-order sensitivity in Eq. (38) can be propagated to the representation space through the margin function. In particular, when $\partial^2 \mathcal{L}_{\text{seg}}/\partial m^2$ becomes large in low-margin regions, the first term in Eq. (45) can amplify local representation-level sensitivity along the direction $\nabla_f m$. Therefore, our representation-level stabilizer provides a practical way to suppress sensitive feature responses induced by low-margin competition, without requiring explicit estimation of the full second-order geometry of the decision boundary.

## Appendix B: Detailed Derivation of Property 2 (Small Stability Radius in Low-Margin Regions)

**Setting and notation.** Recall the logit margin

$$m(x; \theta) \triangleq s_y(x; \theta) - s_{c^*(x)}(x; \theta), \tag{46}$$

where $y$ denotes the target class index, and $c^*(x) = \arg\max_{j \neq y^*} s_j(x; \theta)$. Define the *margin gradient* w.r.t. model parameters as

$$g_m(x) \triangleq \nabla_\theta m(x; \theta). \tag{47}$$

### B.1 Stability Radius as the Minimum Perturbation to Flip the Margin Sign

We quantify local stability around $\theta$ by the smallest parameter perturbation that can change the sign of the margin, i.e., cross the decision boundary between $y$ and $c^*(x)$. Formally, define the (local) stability radius

$$r(x; \theta) \triangleq \inf_{\Delta\theta} \|\Delta\theta\| \quad \text{s.t.} \quad m(x; \theta) \, m(x; \theta + \Delta\theta) \leq 0. \tag{48}$$

The constraint $m(x; \theta) \, m(x; \theta + \Delta\theta) \leq 0$ indicates that the margin becomes zero or changes sign after perturbation, implying that the preference between the two competing logits can flip.

### B.2 First-Order Approximation of Margin Variation

Using the first-order Taylor expansion at $\theta$,

$$m(x; \theta + \Delta\theta) = m(x; \theta) + g_m(x)^\top \Delta\theta + \mathcal{O}(\|\Delta\theta\|^2). \tag{49}$$

When $\|\Delta\theta\|$ is sufficiently small, the higher-order term can be neglected, yielding

$$m(x; \theta + \Delta\theta) \approx m(x; \theta) + g_m(x)^\top \Delta\theta. \tag{50}$$

### B.3 Perturbation Lower Bound for Reaching the Decision Boundary

Crossing the boundary requires a sign change of the margin. Since $m(x; \theta)$ is continuous in $\theta$, any sign change along the segment $\theta(\alpha) = \theta + \alpha\Delta\theta$, $\alpha \in [0, 1]$, must *touch* the boundary $m = 0$ at some intermediate point. Under the first-order model, this boundary-touch condition is approximated by

$$m(x; \theta) + g_m(x)^\top \Delta\theta = 0. \tag{51}$$

Rearranging yields

$$g_m(x)^\top \Delta\theta = -m(x; \theta). \tag{52}$$

Taking absolute values and applying the Cauchy–Schwarz inequality,

$$|m(x; \theta)| = |g_m(x)^\top \Delta\theta| \leq \|g_m(x)\| \, \|\Delta\theta\|. \tag{53}$$

Therefore, any perturbation that reaches the decision boundary must satisfy

$$\|\Delta\theta\| \geq \frac{|m(x; \theta)|}{\|g_m(x)\|}, \quad \text{(i.e., equivalently,} \quad \|\Delta\theta\| \geq \frac{|m(x; \theta)|}{\|\nabla_\theta m(x; \theta)\|}). \tag{54}$$

*Table C.1.* Performance comparison under the disjoint setting on the Pascal VOC 2012 dataset in the 15–5 scenario. The disjoint setting excludes future classes from the background. The best and second-best results are highlighted in **bold** and underlined, respectively.

| Method | Publication | Backbone | Old | New | All |
|--------|-------------|----------|-----|-----|-----|
| MIB (Cermelli et al., 2020) | CVPR 2020 | ResNet101 | 71.8 | 43.3 | 64.7 |
| LGKD (Yang et al., 2023) | ICCV 2023 | ResNet101 | 70.6 | 30.9 | 61.1 |
| CoinSeg (Zhang et al., 2023) | ICCV 2023 | ResNet101 | 79.6 | 43.8 | 71.1 |
| MBS (Park et al., 2024) | ECCV 2024 | ViT | 82.7 | 68.6 | 79.3 |
| **LDKA (Ours)** | – | **ViT** | **83.2** | **69.3** | **79.7** |

*Table C.2.* Comparative results on the Pascal VOC dataset under the overlap 5–3 setting. **Bold** and underlined denote the best and second-best results, respectively. The † symbol indicates results reproduced using the same version of ViT. In addition to the five overlap configurations extensively evaluated in the main paper, we further report the 5–3 setting, which starts from fewer initial classes and is therefore more challenging.

| Method | Backbone | 5-3 (6 steps) | | |
|--------|----------|---------------|---|---|
| | | 0-5 | 6-20 | All |
| GSC (Cong et al., 2023) | ResNet101 | 32.7 | 30.1 | 30.9 |
| EWF (Xiao et al., 2023) | ResNet101 | 61.7 | 42.2 | 47.7 |
| RCIL (Zhang et al., 2022a) | ResNet101 | 65.3 | 41.5 | 50.3 |
| IDEC (Zhao et al., 2023) | ResNet101 | 67.1 | 49.0 | 54.1 |
| GS$^2$K (Cong et al., 2024) | ResNet101 | 58.4 | 53.4 | 54.8 |
| CoGaMiD (Zhu et al., 2026) | ResNet101 | 73.7 | 63.1 | 66.1 |
| MIB[†] (Cermelli et al., 2020) | ViT | 55.2 | 48.9 | 50.7 |
| SSUL[†] (Cha et al., 2021) | ViT | 72.8 | 51.2 | 57.4 |
| MicroSeg[†] (Zhang et al., 2022b) | ViT | 77.8 | 60.3 | 65.3 |
| Coinseg[†] (Zhang et al., 2023) | ViT | 76.1 | 65.4 | 68.5 |
| STAR[†] (Chen et al., 2023a) | ViT | 76.6 | 68.2 | 70.6 |
| CoGaMiD (Zhu et al., 2026) | ViT | **79.9** | 72.7 | 74.7 |
| **LDKA (Ours)** | **ViT** | 79.5 | **75.6** | **76.7** |

## B.4 Interpretation: Why Low-Margin Regions Have a Small Stability Radius

Eq. (54) shows that the minimal perturbation magnitude required to reach (or cross) the boundary scales with $|m(x; \theta)|$. In low-margin regions, the top-1 and top-2 logits are close, hence

$$|m(x; \theta)| \approx 0, \tag{55}$$

which implies a small stability radius:

$$r(x; \theta) \approx \frac{|m(x; \theta)|}{\|\nabla_\theta m(x; \theta)\|}. \tag{56}$$

Consequently, low-margin pixels are *fragile*: even a small parameter update can flip the sign of the margin and alter the local decision between $y$ and $c^*(x)$. This fragility is particularly detrimental in class-incremental learning, where updates are constrained by stability objectives and can readily induce unintended boundary shifts around these low-margin regions.

## Appendix C: Additional Experiments.

### C.1 Performance Comparison in the Disjoint Setting.

In addition to the commonly adopted overlap setting, we further report results under the disjoint setting on the Pascal VOC 2012 dataset. Different from the overlap setting, the disjoint setting excludes future classes from the background category. Specifically, at each incremental step, the background only contains pixels belonging to previously learned classes and the classes introduced at the current step.

*Table C.3.* Results of the proposed method with a CNN-based ResNet-101 backbone under the Pascal VOC 19–1 setting. The results show that our method remains effective beyond the ViT-based instantiation used in the main paper.

| Class | 19–1 (2 steps) | | |
|---|---|---|---|
| | MIoU | Acc | Rec |
| Background | 92.8 | 96.1 | 96.3 |
| Aeroplane | 89.9 | 96.3 | 93.1 |
| Bicycle | 39.2 | 91.0 | 40.8 |
| Bird | 91.4 | 96.5 | 94.6 |
| Boat | 70.0 | 91.4 | 74.9 |
| Bottle | 77.9 | 92.9 | 82.9 |
| Bus | 95.0 | 97.2 | 97.7 |
| Car | 86.0 | 94.0 | 91.0 |
| Cat | 94.1 | 98.6 | 95.3 |
| Chair | 31.9 | 40.9 | 58.9 |
| Cow | 83.4 | 94.0 | 88.0 |
| Diningtable | 54.8 | 61.7 | 83.1 |
| Dog | 89.0 | 94.9 | 93.5 |
| Horse | 79.9 | 88.2 | 89.4 |
| Motorbike | 88.6 | 95.9 | 92.0 |
| Person | 88.6 | 92.3 | 95.8 |
| Potted plant | 58.2 | 70.8 | 76.6 |
| Sheep | 76.4 | 88.7 | 84.7 |
| Sofa | 47.8 | 55.7 | 77.3 |
| Train | 86.7 | 93.9 | 91.9 |
| Tv monitor | 59.0 | 83.5 | 66.8 |
| **Overall** | **75.3** | **86.6** | **83.9** |

As shown in Table C.1, we compare our method with representative incremental segmentation approaches under the 15–5 disjoint configuration, where 15 classes are treated as old classes and the remaining 5 classes are introduced as new classes. Despite the stricter supervision protocol, our method demonstrates a favorable balance between stability and plasticity. In particular, it achieves competitive performance on old classes while maintaining strong segmentation accuracy on newly introduced classes, resulting in the best overall performance among the compared methods under the disjoint setting.

Compared with MBS, our method achieves consistent improvements across all evaluation metrics, including a gain of 0.5% on old classes, 0.7% on new classes, and 0.4% in overall mIoU. Moreover, our approach substantially outperforms earlier methods such as MIB, LGKD, and CoinSeg on new classes with substantial improvements, highlighting its superior capability in learning newly introduced categories under the disjoint setting.

### C.2 Additional Results on the Overlap 5-3 Configuration

In addition to the five overlap incremental configurations (15–5, 19–1, 15–1, 2–2, and 10–1) extensively evaluated in the main paper, we further report results under the overlap 5–3 configuration. Here, "overlap" means that images at each incremental step may contain both previously learned and newly introduced classes. Compared with 15–1, the 5–3 setting starts from fewer initial classes while keeping the same number of incremental steps, which makes the long-horizon learning process more challenging.

As shown in Table C.2, our method achieves the best overall performance under this setting. In particular, compared with the strongest baseline, the overall mIoU improves from 74.7 to 76.7, while the new-class mIoU improves from 72.7 to 75.6. These results indicate that LDKA remains effective even when the initial category set is smaller and the subsequent incremental process is more demanding.

*Table C.4.* Results on a SegFormer-based architecture under the Pascal VOC 15–5 setting. Our method achieves the best overall performance and the best new-class performance, further supporting that the gain is not tied to a specific backbone or decoder form.

| Method | 0-15 | 16-20 | All |
|---|---|---|---|
| ILT (Michieli & Zanuttigh, 2019) | 49.1 | 54.0 | 50.3 |
| MiB (Cermelli et al., 2020) | 78.8 | 60.9 | 74.5 |
| PLOP (Douillard et al., 2021) | 72.5 | 48.4 | 66.8 |
| SATS (Qiu et al., 2023) | 80.2 | 61.2 | 75.7 |
| Ours | **79.7** | **66.6** | **76.6** |

*Table C.5.* Computation cost analysis. Compared with the recent baseline, our approach yields clear performance gains while maintaining similar computational cost. This highlights a favorable balance between efficiency and performance in class-incremental semantic segmentation.

| Method | Params (M) | GFLOPs | Train iter time (ms/iter) | Peak train GPU memory (GiB) | Inference time (ms/img) |
|---|---|---|---|---|---|
| MBS | 102.41 | 103.15 | 171.4 | 9.42 | 15.0 |
| LDKA (Ours) | 102.41 | 103.15 | 179.9 | 9.65 | 15.1 |

## C.3 Experiments with Different Backbones

In class-incremental semantic segmentation (CISS), mainstream methods typically focus on designing effective incremental learning mechanisms under a fixed segmentation architecture, rather than redesigning backbone-specific modules. Accordingly, our goal here is to verify whether the proposed optimization design remains effective when instantiated on different representative backbone families, instead of being tied to a specific backbone form.

In the main paper, all compared methods use the same ViT-based architecture and training protocol, so the reported differences are intended to reflect the effectiveness of the incremental learning strategy itself. To further examine backbone generality beyond ViT, we additionally evaluate our method in two representative non-ViT settings: (i) a classical CNN-based architecture, i.e., DeepLab-v3 with a ResNet-101 backbone, and (ii) a SegFormer-based segmentation framework.

**ResNet-101 backbone.** For the ResNet-101 setting, we follow the standard DeepLab-v3-based CISS protocol and evaluate under the 19–1 incremental configuration. Importantly, MAD is not inherently tied to Transformer class tokens. In the CNN setting, we instantiate the corresponding class representations using the segmentation classifier weights, so that the same class-competition-guided decoupling objective can still be applied. The quantitative results are shown in Table C.3. Under the same ResNet-101 architecture, our method achieves an overall mIoU of 75.3, improving over CoMasTRe by 0.5 mIoU. This suggests that the gain of the proposed method is not restricted to a ViT-specific instantiation.

**SegFormer-based framework.** Following the reviewer's suggestion, we further test LDKA in a SegFormer-based segmentation framework under the Pascal VOC 15–5 setting. The results are reported in Table C.4. Compared with representative replay-free CISS baselines, our method achieves the best overall mIoU (76.6) and the best new-class mIoU (66.6). In particular, compared with the second-best method SATS, our method improves new-class performance by 5.4 mIoU while maintaining competitive old-class performance. This further indicates that the observed gain arises from the proposed optimization design, rather than from a particular backbone or decoder form.

Overall, the above results on ViT (main paper), ResNet-101, and SegFormer consistently support that the proposed method is compatible with different segmentation backbones and architectures. The implementation form of MAD changes with the carrier of class representations (e.g., decoder tokens in ViT-style models and classifier weights in CNN-style models), but the underlying class-decoupling principle remains the same.

## C.4 Computation Cost

We further evaluate the computational cost of the proposed method in terms of parameter size, FLOPs, training iteration time, peak GPU memory usage, and inference latency. All measurements are conducted on a single NVIDIA GeForce RTX 3090 GPU with a batch size of 2 and an input resolution of $512 \times 512$. We use PyTorch 1.10.1 (CUDA 11.1) with automatic mixed precision (AMP) enabled for all compared methods to ensure a fair and consistent runtime environment.

*Table C.6.* Analysis of the ratio $\lambda$ between stability slack $|m(x;\theta)|$ and sensitivity cost $\kappa(x;\theta)$. A balanced ratio achieves the best performance.

| Num | $\lambda$ | 15–1 (6 steps) | | |
| --- | --- | --- | --- | --- |
| | | 0–15 | 16–20 | All |
| 1 | 0.5 | 83.2 | 68.8 | 79.8 |
| **2** | **1.0** | **83.3** | **69.9** | **80.1** |
| 3 | 1.5 | 82.5 | 68.1 | 79.1 |

*Figure C.7.* Analysis of the loss weights in LDKA. As the weight of $\mathcal{L}_{\text{seg-PML}}$ increases, the overall mIoU steadily improves and reaches the optimum at $\alpha = 1$. The performance of $\mathcal{L}_{\text{SKD}}$ remains stable when $\beta \in [0.4, 1.0]$, while $\mathcal{L}_{\text{MAD}}$ performs stably when $\gamma \in [0.6, 1.0]$, with the best result at $\gamma = 1.0$. These results support the default setting $\alpha = \beta = \gamma = 1$.

As reported in Table C.5, MBS and our method share the same architecture, resulting in identical model complexity in terms of parameter count (102.41M) and GFLOPs (103.15). This confirms that the performance gain is not due to additional model capacity. Importantly, the inference latency remains nearly unchanged (about 15 ms per image), indicating negligible deployment-time overhead.

Regarding training efficiency, the per-iteration time increases slightly from 171.4 ms/iter for MBS to 179.9 ms/iter for our method, i.e., an additional 8.5 ms/iter. Peak GPU memory also increases only modestly, from 9.42 GiB to 9.65 GiB. Although our method introduces margin-directional second-order sensitivity stabilization terms, the practical overhead remains modest. Overall, these results indicate that LDKA maintains comparable computational efficiency while achieving stronger class-incremental segmentation performance.

**C.5 Ratio Analysis for PML**

To further analyze the effect of the ratio between $|m(x;\theta)|$ and $\kappa(x;\theta)$, we conduct experiments on Pascal VOC under the 15–1 configuration using three settings of $\lambda$. As shown in Table C.6, the best performance is achieved when the ratio is balanced, i.e., $\lambda = 1.0$, which is also the default choice used in the main paper.

When either term is overemphasized ($\lambda = 0.5$ or $\lambda = 1.5$), the performance becomes weaker than that of the balanced setting. This suggests that the two terms are complementary: they are aligned in optimization goal but capture stability and sensitivity from different perspectives. Overall, the results support the rationality of using the balanced setting in our final

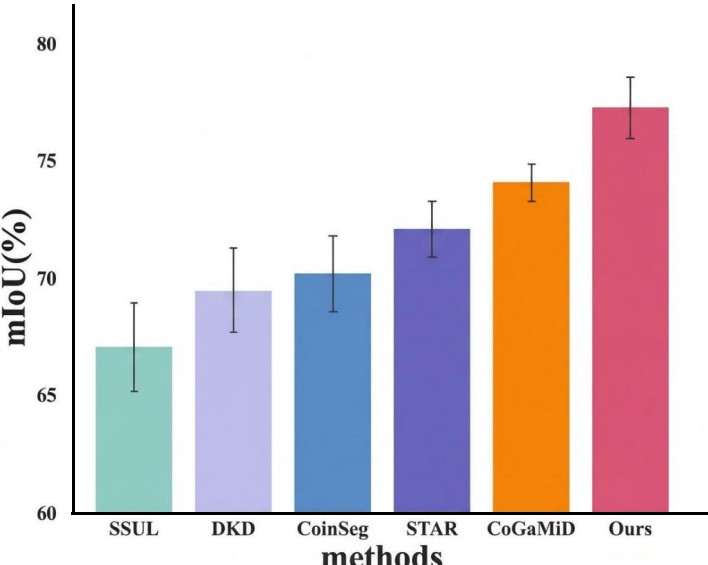

*Figure C.8.* Average performance across 20 different incremental class orders. Our method achieves the best average mIoU with a smaller standard deviation, indicating stronger robustness to class-order variation.

model.

## C.6 Loss Proportion Analysis

In the main paper, the overall loss consists of three components: $\mathcal{L}_{\text{seg-PML}}$, $\mathcal{L}_{\text{SKD}}$, and $\mathcal{L}_{\text{MAD}}$. Throughout all experiments, we use the default setting in which all three weights are set to 1. To further examine the sensitivity of the method to these loss proportions, we conduct additional experiments on the 15–1 setting.

As shown in Fig. C.7, the performance associated with $\mathcal{L}_{\text{seg-PML}}$ steadily improves as $\alpha$ increases and reaches its optimum at $\alpha = 1$. For $\mathcal{L}_{\text{SKD}}$, the performance remains stable for $\beta$ in the range $[0.4, 1.0]$. For $\mathcal{L}_{\text{MAD}}$, the performance also remains stable in the range $[0.6, 1.0]$, with the best result achieved at $\gamma = 1.0$. These observations support the reasonableness of using the default setting $\alpha = \beta = \gamma = 1$.

## C.7 Class Order Sensitivity

Following previous works (Zhang et al., 2023; Zhu et al., 2026), we evaluate our method across 20 different incremental class orders and report both the mean and standard deviation. These experiments examine robustness to class-order variation, which is important in class-incremental learning because the order in which classes are introduced can affect the learning trajectory.

As shown in Fig. C.8, our method achieves the best average performance across all categories while also exhibiting a smaller standard deviation than competing approaches, including SSUL (Cha et al., 2021), DKD (Baek et al., 2022), CoinSeg (Zhang et al., 2023), STAR (Chen et al., 2023a), and CoGaMiD (Zhu et al., 2026). This indicates that our method not only performs strongly on average, but is also more stable under different incremental orders.

## C.8 Clarification of the Novelty of PML and Comparison with Curriculum Learning

In this section, we clarify that our Progressive Margin Learning (PML) method is fundamentally distinct from curriculum learning. Although PML may appear similar to curriculum learning due to its progressive training approach, we emphasize that PML is theoretically different from curriculum learning in both its optimization objectives and control mechanisms. The novelty of PML lies in its ability to focus on low-margin regions, a challenge not addressed by traditional curriculum learning.

Curriculum learning (Bengio et al., 2009) is traditionally defined as a training paradigm where samples are ordered according to a predefined or estimated difficulty measure, and the model is gradually exposed to harder samples following a simple-

to-hard schedule. The core assumption is that sample difficulty is an intrinsic property of the data, and the curriculum is typically implemented as a sample selection or re-weighting strategy over the training set. Importantly, curriculum learning focuses on selecting samples based on their inherent complexity, progressively leading the model through easier to more challenging examples. In contrast, PML does not operate on sample difficulty nor does it perform sample selection. Instead, PML is explicitly designed to address optimization instability in low-margin regions. The progression in PML is governed by the model-dependent margin, which reflects local decision uncertainty rather than data complexity. Importantly, this margin evolves dynamically with model parameters and is not a static property of the input sample. PML focuses on stabilizing optimization by directing the model's attention to the regions with the greatest instability (low-margin), which are often the most challenging in incremental learning settings. From an optimization perspective, PML does not prioritize "easy samples" as curriculum learning does. Instead, it reallocates the optimization budget by smoothly shifting gradient emphasis from high-margin (stable) regions to low-margin (unstable) regions, based on the model's learning progress. All samples are involved in training, but their relative contribution to the loss is continuously adjusted according to margin-induced learnability.

**Key Theoretical Differences.** The distinction between curriculum learning and PML can be summarized as follows:

- **Control Variable:** Curriculum learning controls training via sample difficulty, whereas PML controls optimization through margin-induced stability, focusing on the model's uncertainty rather than the sample difficulty.

- **Optimization Objective:** Curriculum learning aims to ease optimization by ordering samples, while PML aims to stabilize optimization dynamics near decision boundaries, particularly in low-margin regions.

- **Dynamics:** Curriculum schedules are typically predefined or heuristic, whereas PML is adaptive and fully model-driven, dynamically adjusting based on the learning progress.

- **Sample Emphasis:** While curriculum learning uses all samples, it prioritizes easier samples early in training and gradually introduces harder ones. In contrast, PML does not reorder samples but dynamically adjusts the optimization focus across all samples based on margin-related uncertainty, continuously refining low-margin regions where the model is uncertain.

Therefore, PML should not be viewed as a variant of curriculum learning. Rather, it constitutes an optimization-driven learning strategy that specifically targets the intrinsic instability of low-margin regions, a challenge that traditional curriculum learning does not address. The innovation of PML lies in its ability to stabilize learning in low-margin regions, a critical issue in class-incremental semantic segmentation.

**Experimental Validation.** To demonstrate that PML is not merely a curriculum learning strategy, we conducted an experiment where we replaced $\mathcal{L}_{PML}$ with a curriculum learning-based method (Bhat et al., 2021)(It is worth noting that curriculum learning has not been directly applied to CISS). Specifically, we adopted a simple-to-hard approach, where the optimization budget was progressively allocated based on the difficulty of examples, similar to traditional curriculum learning. The experiment was conducted on the 15-1 configuration, the performance with $\mathcal{L}_{PML}$ significantly outperforms the curriculum-based approach, with old-class mIoU improving from 81.4% to 83.3%, and new-class mIoU rising from 66.1% to 69.9%.

This experiment confirms that while both strategies involve a form of progressive learning, the design and impact of PML are distinct and more effective for addressing the specific challenge of low-margin optimization in class-incremental semantic segmentation.

### C.9 Theoretical and Experimental Comparison between SKD and Standard KD

In this section, we analyze the key differences and innovations between our Smooth Knowledge Distillation (SKD) and the traditional Knowledge Distillation (KD), both of which are used to mitigate catastrophic forgetting in class-incremental semantic segmentation. While both approaches share the core concept of transferring knowledge from a teacher model to a student model, the mechanisms by which they achieve this goal differ significantly.

Standard KD (Hinton et al., 2015; Rebuffi et al., 2017) is a method where the soft target probabilities produced by a pre-trained teacher model are used to guide the training of a student model. This process helps the student model retain the

---

**Algorithm 1** Perturbation Stabilization in Smooth Knowledge Distillation

---

**Input:** Current intermediate feature $f^t(x)$, previous-step response $z^{t-1}(x)$, decoder mapping $H_\phi(\cdot)$.
**Output:** Perturbation stabilization loss $\mathcal{L}_{\text{pert}}$.

**Step 1.** Compute the current propagated response:

$$z^t(x) \leftarrow H_\phi\big(f^t(x)\big).$$

**Step 2.** Compute the distillation discrepancy:

$$\mathcal{D}_{\text{kd}}(x) \leftarrow \big\|H_\phi\big(f^t(x)\big) - z^{t-1}(x)\big\|_2^2.$$

**Step 3.** Construct the feature-space perturbation direction:

$$d(x) \leftarrow \nabla_{f^t(x)}\mathcal{D}_{\text{kd}}(x).$$

**Step 4.** Normalize the perturbation direction:

$$\delta^t(x) \leftarrow \frac{d(x)}{\|d(x)\|_2}.$$

**Step 5.** Inject the perturbation into the intermediate feature and propagate it:

$$\widetilde{z}^t(x) \leftarrow H_\phi\big(f^t(x) + \delta^t(x)\big).$$

**Step 6.** Compute the perturbation stabilization loss:

$$\mathcal{L}_{\text{pert}} \leftarrow \mathbb{E}_x\left[\big\|z^t(x) - \widetilde{z}^t(x)\big\|_2^2\right].$$

**return** $\mathcal{L}_{\text{pert}}$.

---

knowledge learned by the teacher, reducing the likelihood of forgetting previously learned classes. However, KD is primarily focused on feature-level distillation, and it does not directly address the issue of optimization instability near decision boundaries, particularly in low-margin regions where the model's confidence is low. In contrast, SKD introduces a significant innovation by incorporating margin-directional second-order sensitivity damping and perturbation-based stabilization during the distillation process. These two stabilizing factors are designed to address the issue of optimization instability near the decision boundary, which is especially problematic in class-incremental learning settings. Specifically, SKD applies a damping term that suppresses excessive second-order sensitivity and stabilizes the learning process, preventing the model from becoming overly sensitive to small perturbations in the feature space, especially in low-margin regions. This leads to a more stable distillation process, enhancing the model's ability to preserve old knowledge while learning new classes.

**Key Theoretical Differences:** The main differences between SKD and standard KD can be summarized as follows:

- **Stability Focus:** While standard KD focuses on transferring knowledge via soft target probabilities, SKD emphasizes stabilizing the learning process by addressing second-order sensitivity and maintaining consistency under perturbations, particularly in low-margin regions.

- **Margin-Directional Second-Order Sensitivity Damping:** SKD introduces a damping term that reduces excessive sensitivity near decision boundaries, which is not present in standard KD.

- **Perturbation-based Stabilization:** SKD introduces a perturbation stabilization mechanism that enhances model robustness during training and reduces the impact of small variations across steps.

**Experimental Validation.** To demonstrate the effectiveness of SKD, we conducted experiments under 15-1 where we replaced SKD with standard KD. The performance of the model with SKD outperforms the version with standard KD, with old-class mIoU improving from 80.2% to 83.3%. This substantial improvement in both old-class and new-class performance demonstrates that SKD's additional stabilization mechanisms significantly enhance the model's ability to retain

---

**Algorithm 2** Practical Computation of Directional Second-Order Sensitivity Damping in SKD

---

**Input:** Current-step decoded feature $z^t(x)$, frozen previous-step decoded feature $z^{t-1}(x)$.
**Output:** Directional second-order sensitivity damping loss $\mathcal{L}_{\text{dss}}$.

**Step 1.** Compute the feature-distillation discrepancy:

$$\mathcal{D}_{\text{fd}}(x) \leftarrow \left\| z^t(x) - z^{t-1}(x) \right\|_2^2.$$

**Step 2.** Compute the first-order gradient with respect to the current-step decoded feature while retaining the computation graph:

$$g(x) \leftarrow \nabla_{z^t(x)} \mathcal{D}_{\text{fd}}(x).$$

**Step 3.** Regularize the squared gradient norm:

$$\mathcal{L}_{\text{dss}} \leftarrow \mathbb{E}_x \left[ \|g(x)\|_2^2 \right].$$

**Step 4.** Backpropagate the surrogate loss through the retained computation graph:

$$\nabla_\theta \mathcal{L}_{\text{dss}} = \nabla_\theta \mathbb{E}_x \left[ \left\| \nabla_{z^t(x)} \left\| z^t(x) - z^{t-1}(x) \right\|_2^2 \right\|_2^2 \right].$$

**return** $\mathcal{L}_{\text{dss}}$.

---

old knowledge while learning new classes, compared to the traditional KD approach. These experimental results further support the effectiveness of SKD, showing that margin-directional second-order sensitivity damping and perturbation-based stabilization jointly help reduce optimization instability and improve overall class-incremental segmentation performance.

**C.10 Algorithm of Perturbation Stabilization**

Algorithm 1 provides the detailed procedure of the perturbation stabilization term in Smooth Knowledge Distillation. The perturbation is constructed in the intermediate feature space and then propagated through the decoder mapping $H_\phi(\cdot)$. Therefore, the perturbed response is defined as $\widetilde{z}^t(x) = H_\phi(f^t(x) + \delta^t(x))$, rather than $z^t(x) + \delta^t(x)$.

This formulation regularizes the downstream propagated response induced by a local feature-space perturbation. Since the perturbation is injected before the decoder mapping, $\mathcal{L}_{\text{pert}}$ does not reduce to the constant-norm term $\mathbb{E}_x \|\delta^t(x)\|_2^2$. Instead, it penalizes unstable response variations caused by local perturbations around the current intermediate representation, which helps improve the stability of low-margin regions during incremental learning.

**C.11 Practical Computation of Directional Second-Order Sensitivity Damping**

This section further details the practical computation of the directional second-order sensitivity damping term in Smooth Knowledge Distillation, as summarized in Algorithm 2. Since explicitly materializing the full Hessian matrix is computationally prohibitive, we instead compute the first-order gradient of the feature-distillation discrepancy with respect to the current-step decoded feature, retain the computation graph, and regularize the squared norm of this gradient. Optimizing the resulting surrogate via double backpropagation provides a tractable approximation to directional second-order sensitivity regularization. The above procedure avoids explicitly constructing the full Hessian matrix. Since the optimization of $\mathcal{L}_{\text{dss}}$ differentiates through the first-order gradient $g(x)$, it induces double backpropagation and provides a tractable approximation to directional second-order sensitivity damping on the decoded feature used for cross-step distillation.

**C.12 Additional Analyses of Perturbation and Related Variants**

This section provides additional analyses of two design choices in LDKA: the perturbation strategy in SKD and the progressive weighting strategy in PML. All experiments are conducted on the Pascal VOC 2012 dataset under the 19–1 incremental setting. The compared variants use the same backbone, decoder, and training protocol as our final model.

*Table C.7.* Analysis of perturbation variants in SKD on Pascal VOC 2012 under the 19–1 setting. The best results are highlighted in **bold**.

| Method | 0–19 | 20 | All |
|---|---|---|---|
| Multi-directional perturbation | 83.0 | 69.4 | 82.4 |
| Adaptive perturbation | **83.2** | 67.0 | 82.4 |
| LDKA (Ours) | 82.9 | **73.9** | **82.5** |

*Table C.8.* Comparison between PML and adaptive curriculum variants on Pascal VOC 2012 under the 19–1 setting. The best results are highlighted in **bold**.

| Method | 0–19 | 20 | All |
|---|---|---|---|
| MentorNet | **83.2** | 70.8 | 82.4 |
| SPL | 83.0 | 70.6 | 82.4 |
| LDKA (Ours) | 82.9 | **73.9** | **82.5** |

**Perturbation variants in SKD.**   We first evaluate two additional perturbation variants for the perturbation stabilization term in SKD. The first variant is a multi-directional perturbation strategy, where two auxiliary directions are introduced in addition to the original normalized gradient direction. These auxiliary directions are orthogonalized against the original direction and against each other, and the same propagation-based consistency constraint is applied to the resulting perturbed feature. The second variant is an adaptive perturbation strategy, where the original perturbation direction is retained, but the perturbation magnitude is scaled according to the severity of the low-margin condition, so that more fragile low-margin samples receive stronger perturbations. Both variants are only used during training and introduce no extra trainable modules or inference-time overhead.

As shown in Table C.7, the two perturbation variants slightly improve old-class performance, but they do not improve the new-class or overall performance over the final design. In particular, LDKA improves the new-class mIoU by 4.5 points over the strongest perturbation variant and achieves the best overall mIoU. This suggests that the original perturbation already provides a favorable stability–plasticity trade-off in this setting. More complex perturbation directions or adaptive magnitudes strengthen old-class stability, but can also weaken the plasticity required for the newly introduced class.

**PML versus adaptive curriculum variants.**   We further compare PML with two representative adaptive curriculum strategies, Self-Paced Learning (SPL) (Kumar et al., 2010) and MentorNet (Jiang et al., 2018). For a fair comparison, only the PML component is replaced, while the backbone, decoder, SKD, MAD, and training protocol remain unchanged. SPL and MentorNet mainly implement difficulty-based adaptive weighting, whereas PML reallocates optimization emphasis according to the model-derived low-margin state and its associated sensitivity.

Table C.8 shows that SPL and MentorNet are competitive on old classes, but they do not surpass PML on new-class learning, which is the key challenge in later incremental steps. Compared with the strongest adaptive curriculum variant, LDKA improves the new-class mIoU by 3.1 points and also achieves the best overall performance. These results suggest that the gain of PML is not merely due to generic adaptive weighting or an easy-to-hard curriculum. Instead, the improvement is more consistently associated with targeted optimization of low-margin regions, where the model faces stronger class competition and higher local sensitivity.

### C.13 Additional Theoretical and Experimental Analysis of MAD.

In class-incremental semantic segmentation (CISS), low-margin regions are where the model faces high uncertainty in its decision boundaries, often leading to instability and catastrophic forgetting. These regions are characterized by the model being less confident about the class predictions, making them particularly susceptible to small perturbations and misclassifications. To address this, we propose Misclassification-Aware Decoupling (MAD), a strategy designed to reduce class interference and enhance the stability of learning in these low-margin regions.

**Theoretical Analysis of MAD and Low-Margin Regions.**   Unlike traditional methods such as contrastive learning, which primarily aim to minimize the distance between class features, MAD targets the inherent instability in low-margin regions. These regions are particularly problematic in incremental learning, where new classes can disturb the decision boundaries of

*Table C.9.* Single-component ablation study on Pascal VOC 2012 under the 15–1 setting. Each component is individually added to the same baseline to evaluate its standalone contribution.

| Method | 0–15 | 16–20 | All |
|---|---|---|---|
| Baseline | 71.8 | 45.0 | 65.4 |
| $+\mathcal{L}_{\text{seg-PML}}$ | **82.2** | **61.4** | **77.2** |
| $+\mathcal{L}_{\text{SKD}}$ | 81.6 | 55.8 | 75.5 |
| $+\mathcal{L}_{\text{MAD}}$ | 81.4 | 59.0 | 76.1 |

old classes. MAD introduces a competition matrix ($\Gamma_{ij}$), which models the misclassification of class pairs. By applying larger penalties to frequently misclassified class pairs, MAD effectively reduces the interference between these classes, particularly in regions where the model's decision boundaries are uncertain.

This approach directly addresses the optimization instability in low-margin regions, where the model faces difficulty distinguishing between classes due to the ambiguous or overlapping decision boundaries. By focusing on the most challenging, often misclassified class pairs, MAD stabilizes the learning process, ensuring that the model can better retain knowledge of old classes while incorporating new ones. The core innovation of MAD lies in its ability to refine the decision boundaries in these unstable regions, improving the overall performance of the model in class-incremental settings.

**Key Differences from Other Methods.** Unlike traditional contrastive learning or orthogonality constraints used for inter-class decoupling, MAD focuses on misclassification-aware optimization. While contrastive learning minimizes the distances between features of the same class and increases the distance between different classes, it does not account for the fact that certain class pairs are more likely to be confused than others. Similarly, orthogonality-based methods, while effective in suppressing semantically related features, often lead to poorer performance due to excessive suppression of useful information. In contrast, MAD specifically identifies and mitigates the impact of misclassified class pairs by introducing the competition matrix, ensuring that the model focuses on resolving the most challenging confusions among classes.

**Experimental Validation.** To evaluate the effectiveness of MAD, we conducted an ablation study in which we compared it against other common class decoupling strategies, including contrastive learning and orthogonality-based methods. The results of the ablation study, as shown in Table 4, demonstrate that MAD outperforms all other strategies. Specifically, incorporating the competition matrix $\Gamma_{ij}$ into contrastive learning improves mIoU for old classes by $0.7\%$ and new classes by $6.5\%$. The orthogonality constraint-based method, on the other hand, suppressed semantically related features, leading to poorer performance. These results confirm that MAD not only reduces class interference but also effectively stabilizes learning in low-margin regions, ensuring that both old and new classes are learned effectively. The ability of MAD to adaptively focus on the most challenging class pairs in low-margin regions makes it a crucial innovation in class-incremental learning.

**C.14 Standalone Contribution of Each Component**

This section further analyzes the standalone contribution of each component in LDKA. In the main paper, Table 3 reports the progressive integration of $\mathcal{L}_{\text{seg-PML}}$, $\mathcal{L}_{\text{SKD}}$, and $\mathcal{L}_{\text{MAD}}$ on the same baseline. Fig. 6 further illustrates their different roles: $\mathcal{L}_{\text{seg-PML}}$ enlarges the top-1/top-2 margin gap and reallocates optimization emphasis toward margin-critical regions; $\mathcal{L}_{\text{SKD}}$ stabilizes sensitive feature responses under local perturbations; and $\mathcal{L}_{\text{MAD}}$ reduces inter-class confusion by decoupling highly competitive class pairs. To make the independent effect of each component more explicit, we further conduct single-component ablations on Pascal VOC 2012 under the 15–1 setting.

As shown in Table C.9, each component brings clear improvements over the baseline when used independently. $\mathcal{L}_{\text{seg-PML}}$ achieves the largest standalone gain, indicating the importance of progressively reallocating optimization emphasis toward low-margin regions. $\mathcal{L}_{\text{SKD}}$ improves both old- and new-class performance by stabilizing the feature-distillation process, while $\mathcal{L}_{\text{MAD}}$ provides a stronger gain on new classes by reducing confusion among competitive class pairs.

These results are consistent with the progressive ablation in the main paper: although each component is effective on its own, their combination remains more effective because the three modules address complementary aspects of the low-margin bottleneck. They form an allocation–stabilization–decoupling strategy for class-incremental semantic segmentation.

## C.15 More Qualitative Analysis.

To further validate the effectiveness of our proposed method in improving pixel-level semantic segmentation accuracy, we present more qualitative comparisons with recent state-of-the-art approaches under the 15-1 setting. Specifically, Figs C.9–C.12 illustrate qualitative results, covering both the old classes from the base stage (0-15) and the new classes from the incremental stage (16-20).

As shown in Fig. C.9, existing methods tend to suffer from background confusion when segmenting previously learned classes. In particular, background regions with similar visual patterns are often misclassified as semantic objects, such as confusing background cloth or windows with *sofa* or *TV monitor*. These results indicate that many methods struggle to suppress spurious activations caused by background shift after learning new classes. In contrast, our method produces cleaner segmentation maps with more accurate object boundaries, demonstrating stronger resistance to background-induced misclassification.

Fig. C.10 presents additional examples where existing methods exhibit severe inter-class confusion among old categories. For instance, visually similar objects such as *bus* and *train*, or *cow* and *sheep*, are frequently confused. Such errors reveal limited capability in preserving fine-grained semantic distinctions during continual learning. Our approach effectively mitigates this issue by maintaining clearer category separation, leading to more consistent and semantically accurate predictions.

In Fig. C.11, we further analyze cases focusing on object shape integrity. Several competing methods fail to retain complete object structures, producing fragmented, over-smoothed, or significantly distorted segmentation results. By contrast, our method better preserves object contours and fine-grained details, resulting in more faithful shape reconstruction for previously learned classes.

Finally, Fig. C.12 shows qualitative comparisons on newly introduced classes. Existing methods often demonstrate limited plasticity when adapting to new semantic concepts, either failing to detect new objects or confusing them with visually similar categories. Moreover, inaccurate object boundaries and background noise are commonly observed. In comparison, our method consistently achieves more accurate segmentation results on new classes, with clearer object boundaries and substantially reduced misclassification. These qualitative results collectively demonstrate that our approach effectively balances stability and plasticity, substantially enhancing pixel-wise classification performance in class-incremental semantic segmentation.

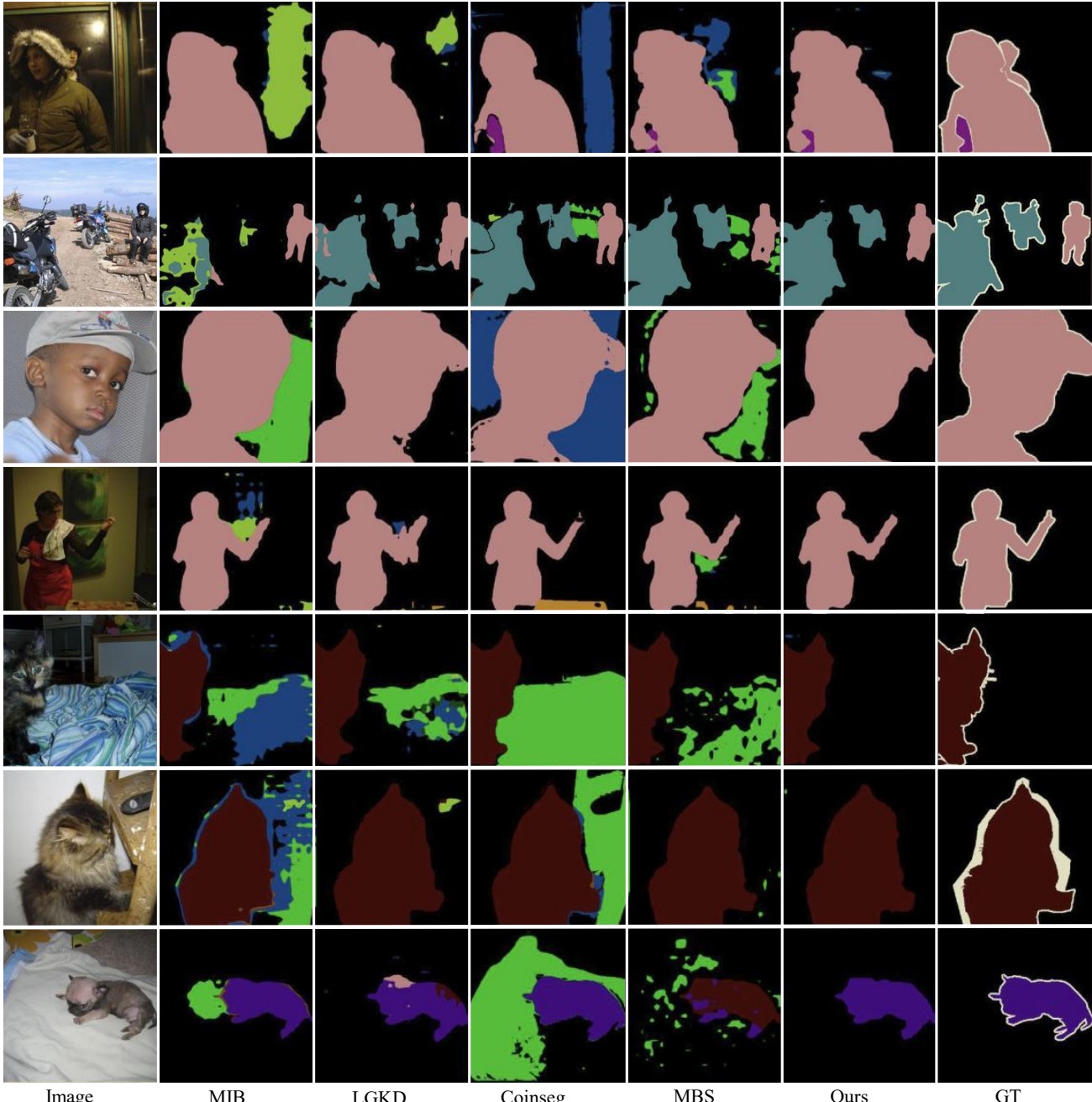

| Image | MIB | LGKD | Coinseg | MBS | Ours | GT |

*Figure C.9.* Qualitative comparison of segmentation results for previously learned classes under the 15–1 setting. Existing methods tend to confuse background regions with semantic objects, leading to background-induced misclassification. Our method produces cleaner predictions with more accurate object boundaries, showing stronger robustness to background shift.

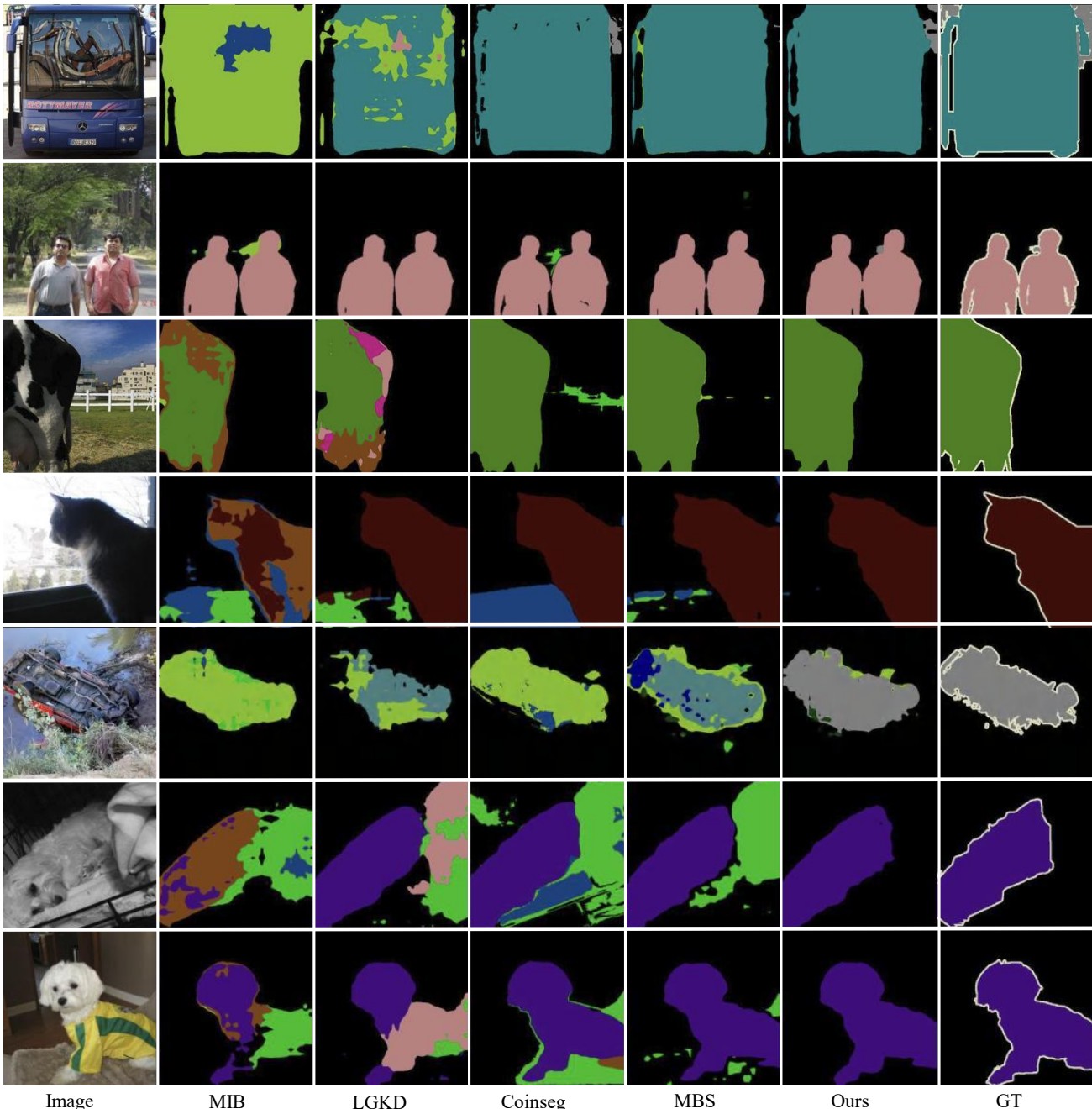

| Image | MIB | LGKD | Coinseg | MBS | Ours | GT |
|-------|-----|------|---------|-----|------|-----|

*Figure C.10.* Qualitative examples illustrating inter-class confusion among old categories. Existing approaches frequently confuse visually similar classes, such as *bus* and *train*, or *cow* and *sheep*. Our method better preserves semantic distinctions, resulting in more consistent and accurate segmentation results.

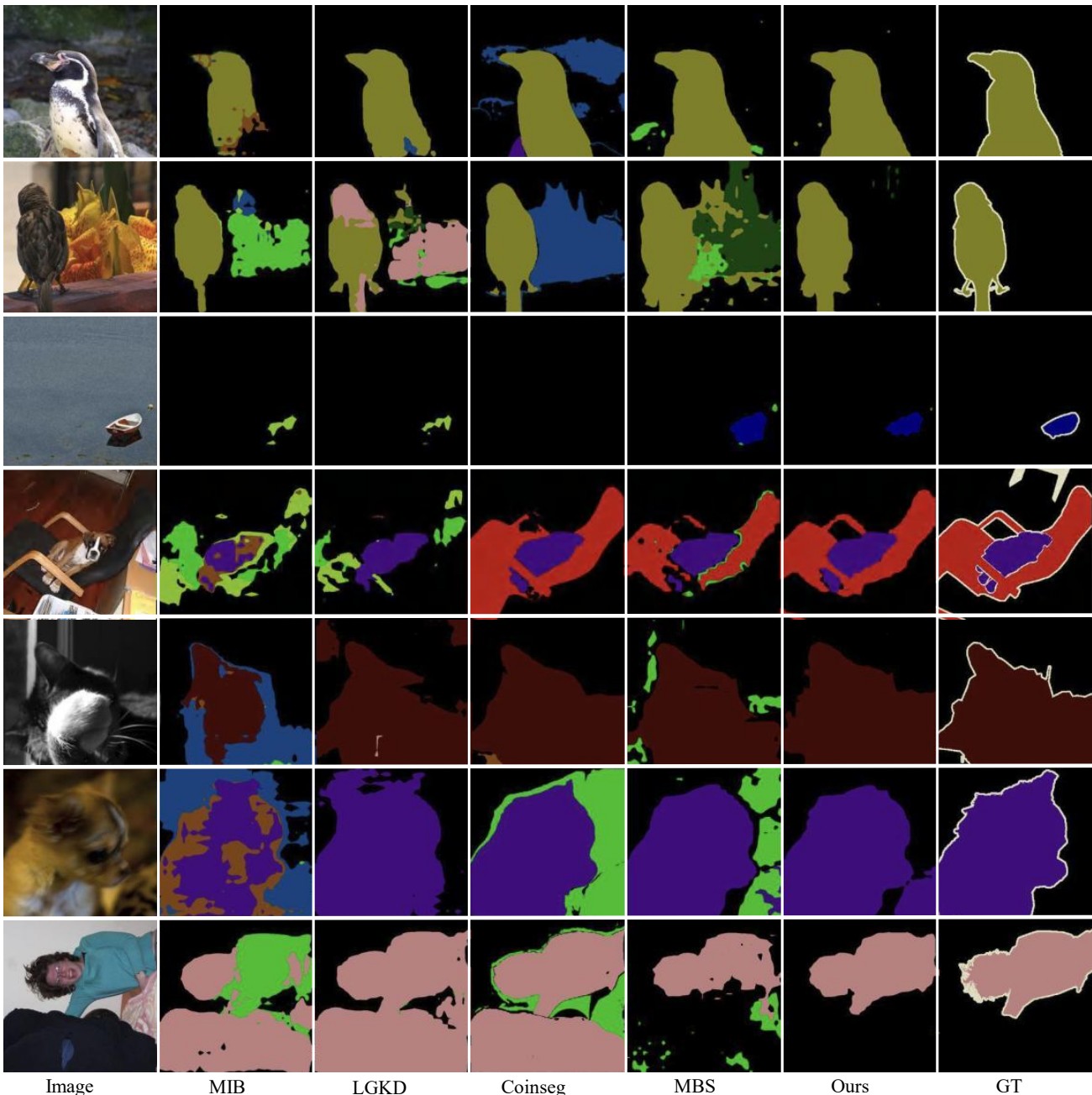

| Image | MIB | LGKD | Coinseg | MBS | Ours | GT |

*Figure C.11.* Qualitative comparison focusing on object shape integrity for previously learned classes. Several methods produce incomplete or distorted object shapes during incremental learning. Our approach preserves finer object structures and contours, leading to improved shape consistency.

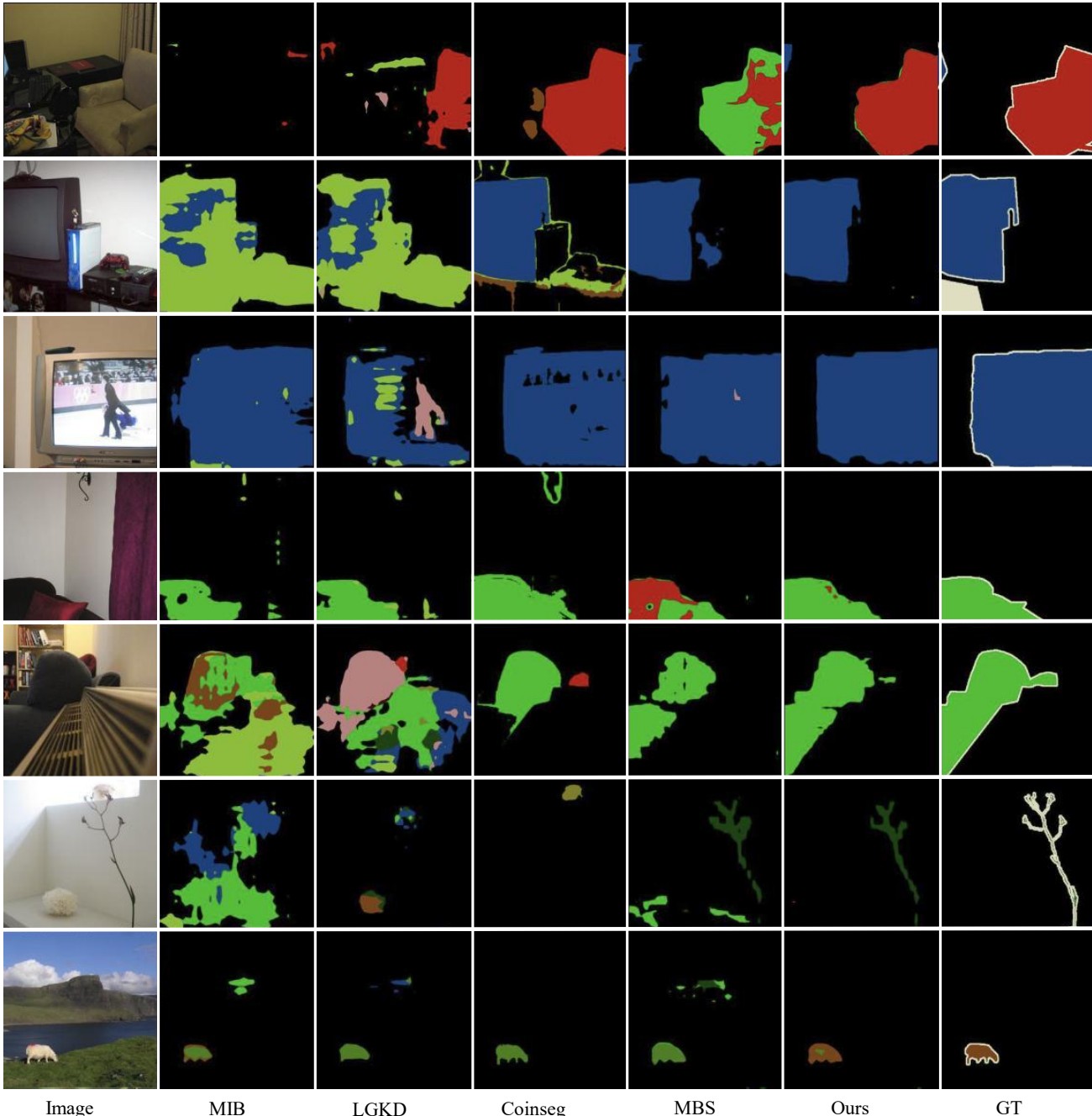

|            |     |      |         |     |      |    |
| ---------- | --- | ---- | ------- | --- | ---- | -- |
| Image | MIB | LGKD | Coinseg | MBS | Ours | GT |

*Figure C.12.* Qualitative comparison on newly introduced classes under the class-incremental setting. Existing methods often fail to accurately segment new categories, suffering from category confusion, shape distortion, or background misclassification. Our method demonstrates superior plasticity by effectively acquiring new semantic concepts with cleaner object boundaries and reduced misclassification.

