# OpenReview forum: "Learnability-Driven Knowledge Assimilation for Class-Incremental Semantic Segmentation"
_ICML.cc/2026/Conference — ICML 2026 regular_

### Official Review · Reviewer_UfuW · 2026-03-06

**Soundness:** 3
**Presentation:** 3
**Significance:** 3
**Originality:** 3
**Overall Recommendation:** 5
**Confidence:** 4

**Summary:**

This paper proposes Learnability-Driven Knowledge Assimilation (LDKA) for class-incremental semantic segmentation (CISS). The authors analyze optimization challenges in low-margin regions and argue that such regions exhibit high curvature and small stability radii, which can negatively affect knowledge transfer during incremental learning. Based on this analysis, the method introduces three components: Progressive Margin Learning (PML), Smooth Knowledge Distillation (SKD), and Misclassification-Aware Decoupling (MAD). These components aim to progressively incorporate difficult samples, stabilize knowledge distillation, and mitigate misclassification effects during incremental training. Experiments under multiple incremental settings demonstrate improved performance compared with existing approaches.

**Compliance With Llm Reviewing Policy:**

Affirmed.

**Final Justification:**

The proposed method has a reasonable theoretical foundation and can achieve competitive performance,the original score is maintained.

**Key Questions For Authors:**

1、To what extent do the performance gains of LDKA depend on the ViT backbone rather than the proposed components (PML, SKD, MAD)? Could the authors provide controlled experiments isolating the contribution of each component?
2、How does PML compare with more advanced adaptive curriculum learning methods, such as Self-Paced Learning or MentorNet, beyond the simple curriculum comparison in Appendix C.8?
3、Can LDKA’s components be applied effectively to other segmentation frameworks, such as CNN-based DeepLab or transformer-based SegFormer? If so, can the authors provide experimental results or discussion?

**Limitations:**

The authors partially discuss limitations, mainly focusing on the method’s evaluation being restricted to ViT-based architectures. However, the paper does not clearly discuss the dependency on the backbone, the potential sensitivity of the method to hyperparameters, or how well the components generalize to other segmentation frameworks. Additionally, there is no mention of potential negative societal impacts of class-incremental segmentation in real-world applications. It is recommended that the authors explicitly address these points, including the dependency on the backbone,component-wise contribution and robustness, and generalization to other architectures.

**Strengths And Weaknesses:**

The paper proposes Learnability-Driven Knowledge Assimilation (LDKA) for class-incremental semantic segmentation, focusing on optimizing learning in low-margin regions that exhibit high curvature and small stability radii. The method introduces three complementary components—Progressive Margin Learning (PML), Smooth Knowledge Distillation (SKD), and Misclassification-Aware Decoupling (MAD)—which are motivated by theoretical analysis and validated through experiments across nine incremental configurations. The paper is generally well-written and structured, with a clear description of each component and its role in improving incremental learning. The experimental results show consistent improvements over existing baselines, demonstrating practical utility. However, some limitations remain. Some of the performance gains may be partially attributed to the use of a ViT backbone rather than the method itself, and controlled experiments isolating the contribution of each component are limited. The distinction between PML and standard curriculum learning strategies, as well as more advanced adaptive approaches (e.g., Self-Paced Learning, MentorNet), could be clarified. Additionally, evaluation is primarily on ViT-based architectures, leaving the generality of the approach to other segmentation frameworks, such as CNN-based DeepLab or transformer-based SegFormer, uncertain. Finally, the novelty mainly comes from the combination and integration of the three components rather than fundamentally new theoretical insights, and further experiments would help quantify the independent contribution of LDKA itself.

---

> ### Author Rebuttal · Authors · 2026-03-27
>
> We sincerely thank the reviewer for the **positive assessment** of our paper, especially for recognizing the **motivation**, the **clear structure of LDKA**, and the **consistent empirical improvements** across multiple incremental settings. We also greatly appreciate the reviewer’s constructive suggestions regarding **backbone dependence**, **component-wise contribution**, **comparison with adaptive curricula**, and **generality to other segmentation frameworks**.
>
> **Q1 [Backbone dependence / component-wise contribution]:** Thank you for this important question. In the main paper, following prior ViT-based CISS practice, we use the **same ViT-B/16 encoder, decoder, and training protocol** for all compared methods. Under this controlled setting, the performance differences are intended to reflect the effectiveness of the **incremental learning strategy**, rather than the mere choice of a ViT backbone.
>
> Regarding **component-wise contribution**, the paper has provided a controlled ablation in **Table 3**, where $L_{\text{seg-PML}}$, $L_{\text{SKD}}$, and $L_{\text{MAD}}$ are introduced progressively on the same baseline. **Sec. 5.3 / Fig. 6** further clarifies their roles: $L_{\text{seg-PML}}$ enlarges the margin gap and shifts optimization toward margin-critical regions; $L_{\text{SKD}}$ suppresses curvature and enlarges the stability radius; and $L_{\text{MAD}}$ reduces inter-class confusion.
>
> To make the standalone contribution of each component more explicit, we further conduct **single-loss ablations** on **Pascal VOC 2012, 15-1**:
>
> | Method | 0-15 | 16-20 | All |
> |---|---:|---:|---:|
> | Baseline | 71.8 | 45.0 | 65.4 |
> | + $L_{\text{seg-PML}}$ | 82.2 | 61.4 | 77.2 |
> | + $L_{\text{SKD}}$ | 81.6 | 55.8 | 75.5 |
> | + $L_{\text{MAD}}$ | 81.4 | 59.0 | 76.1 |
>
> These results are consistent with **Table 3** and **Fig. 6**: each component is effective on its own, while their combination remains best due to their complementary **allocation–stabilization–decoupling** roles.
>
> **Q2 [PML vs. adaptive curriculum variants]:** Thank you for this valuable suggestion. We further replace **PML** with two representative adaptive curricula, **Self-Paced Learning (SPL)** and **MentorNet**, while keeping the **same backbone, decoder, SKD, MAD, and training protocol** unchanged. SPL and MentorNet mainly implement **difficulty-based adaptive weighting**, whereas PML reallocates optimization emphasis according to the **model-derived low-margin state** and its associated sensitivity.
>
> We evaluate these variants on **Pascal VOC 2012, 19-1**:
>
> | Method | 0-19 | 20 | All |
> |---|---:|---:|---:|
> | MentorNet | **83.2** | 70.8 | 82.4 |
> | SPL | 83.0 | 70.6 | 82.4 |
> | Ours | 82.9 | **73.9** | **82.5** |
>
> Although SPL and MentorNet are competitive, they do not surpass PML on **new-class learning**, which is the key challenge in later incremental steps. Our method achieves the best new-class mIoU (+3.1 over the strongest adaptive competitor) and the best overall performance, suggesting that the benefit of PML is more consistent with targeted optimization of low-margin regions than with generic easy-to-hard weighting alone. We have clarified this distinction more explicitly in the paper.
>
> **Q3 [Backbone generality]:** To further examine **generalization** beyond ViT, the paper already includes cross-backbone validation in **Appendix C.3**, where LDKA is evaluated with **DeepLab-v3 + ResNet-101**. Under the same ResNet-101 architecture, LDKA improves the mean mIoU over all categories by 0.5 compared with **CoMasTRe**, indicating that the gain is not tied to a specific ViT instantiation.
>
> Following the reviewer’s suggestion, we further tested LDKA on a **SegFormer-based** architecture under **Pascal VOC 15-5**:
>
> | Method | 0-15 | 16-20 | All |
> |---|---:|---:|---:|
> | ILT | 49.1 | 54.0 | 50.3 |
> | MiB | 78.8 | 60.9 | 74.5 |
> | PLOP | 72.5 | 48.4 | 66.8 |
> | SATS | **80.2** | 61.2 | 75.7|
> | Ours |79.7| **66.6** | **76.6** |
>
> These results further support that the observed gains arise from the proposed **optimization design**, rather than from a particular backbone choice.
>
> **Further discussion on stability and broader impacts:** We thank the reviewer for highlighting these points. **Hyperparameter stability** is evaluated in **Appendix C.5–C.6**, where we analyze the ratio between stability slack and sensitivity cost, as well as the loss weights. These results show that LDKA remains robust across a broad range of settings. **Broader impacts** are discussed in **Sec. 7 (p.9)**. LDKA can benefit applications such as **robot perception** and **autonomous driving**, while our experiments use only **public datasets** and involve no private or sensitive data.
>
> Once again, we sincerely thank the reviewer for the **highly constructive suggestions** and the **positive recognition** of our work.

---

> > ### Author Rebuttal · Reviewer_UfuW · 2026-04-02
> >
> > The proposed method has a reasonable theoretical foundation and can achieve competitive performance. However, compared with some classic algorithms, it only shows comparable advantages rather than clear superiority. It is suggested that the authors further improve the proposed method, including exploring its potential for practical deployment and real‑world applications.

---

> > > ### Author Response · Authors · 2026-04-02
> > >
> > > Dear Reviewer UfuW,
> > >
> > > We would like to express our sincere **gratitude to the Reviewer for the exceptionally professional, insightful, and constructive feedback** throughout the review process. We are particularly encouraged that the Reviewer recognizes that our method has a “**reasonable theoretical foundation**” and “**can achieve competitive performance**”. We are especially grateful for this positive assessment and for the **Reviewer’s recognition that the work has reached the level of an acceptable contribution**.
> > >
> > > Regarding the concern that our method shows comparable advantages rather than clear superiority over some classic algorithms, we **respectfully note** that the strength of our method lies **not only in maintaining old-class performance and mitigating forgetting, but also in achieving more pronounced gains on new classes**. In particular, under the 19-1 setting, for the final incremental class, our method improves the performance of the 20th class by **3.1 points** over the classical methods. We believe this is a meaningful result, because in CISS, improving new-class learning while preserving old-class stability is precisely the central challenge, and **gains on the newly added class are especially indicative of stronger plasticity under the incremental constraint**.
> > >
> > > At the same time, we sincerely appreciate the Reviewer’s valuable forward-looking suggestion to further improve the proposed method, including **exploring its potential for practical deployment and real-world applications**. We fully agree that this is an important and worthwhile direction. In our view, **autonomous driving and robotic perception are particularly meaningful real-world scenarios for this research**, because both are increasingly moving toward open-world operation, where new categories and semantic concepts may continuously appear after a system has already been deployed. This creates **a strong practical need for continual learning capability at the perception level**. In such settings, repeatedly collecting all historical data and conducting full joint retraining is often unrealistic due to storage costs, privacy restrictions, and deployment efficiency requirements. By contrast, our replay-free class-incremental semantic segmentation method is naturally suited to this demand, since it is designed to continually learn newly introduced classes while retaining previously acquired knowledge without requiring access to historical training data. In future work, we will continue to improve the method with these application scenarios in mind, particularly toward more realistic evaluation and deployment in autonomous driving and robotic perception systems under open-world conditions.
> > >
> > > The Reviewer’s comments have not only helped us better understand how the work can be further strengthened, but have also provided meaningful guidance for the next stage of our research. **We are truly grateful for the Reviewer’s high academic standards, professional judgment, and constructive encouragement**.
> > >
> > > With sincere appreciation and respect,
> > > The Authors

---

### Official Review · Reviewer_f11d · 2026-03-10

**Soundness:** 2
**Presentation:** 3
**Significance:** 3
**Originality:** 3
**Overall Recommendation:** 4
**Confidence:** 4

**Summary:**

This paper tackles the core challenge of balancing old-class stability and new-class plasticity in class-incremental semantic segmentation (CISS). While existing methods mitigate catastrophic forgetting, new-class performance is limited by low-margin regions—pixels with near-equal logits for the ground-truth and competing classes, characterized by high loss curvature and small stability radius. The authors outline the central question of how to optimize these ill-conditioned regions, and they claim to identify a pertinent issue: existing methods neglect this optimization bottleneck despite stabilizing old knowledge. To address this, the paper proposes LDKA, a three-component framework (PML for adaptive optimization budget allocation, SKD for curvature damping and perturbation stabilization, MAD for reducing inter-class confusion via a competition matrix). Experiments on Pascal VOC 2012 and ADE20K across 9 protocols show LDKA consistently improves new-class mIoU while preserving old-class performance, outperforming SOTA methods.

**Compliance With Llm Reviewing Policy:**

Affirmed.

**Final Justification:**

The authors provided a detailed rebuttal and additional results that satisfactorily addressed my concerns. The clarifications improved the completeness and overall strength of the paper. I therefore maintain my recommendation of Weak accept.

**Key Questions For Authors:**

Please refer to weaknesses

**Limitations:**

yes

**Strengths And Weaknesses:**

### Strengths:
- The paper provides a rigorous optimization-based analysis of low-margin regions, formally proving their high curvature and small stability radius as core CISS bottlenecks—this theoretical foundation fills a critical gap in existing CISS research that focuses only on forgetting mitigation.
- The paper explicitly distinguishes LDKA’s components from related methods (e.g., PML vs. curriculum learning, SKD vs. standard KD), experimentally validating that its design is not a trivial extension but a targeted solution for CISS-specific low-margin optimization.
- LDKA achieves consistent improvements over recent SOTA methods on both new and old classes, with minimal computational overhead (matching model size/FLOPs of baselines and negligible inference latency), making it deployable for real-world CISS tasks.

### Weaknesses:
- SKD’s perturbation stabilization uses a single normalized gradient direction for feature perturbations; there is no analysis of multi-directional or adaptive perturbations, which may further improve stability in low-margin regions with complex feature distributions.
- The paper relies on pseudo-labels for old-class supervision, but pseudo-labels are inherently noisy. However, there is no systematic analysis of how noise in pseudo-labels propagates to LDKA’s components—e.g., whether incorrect pseudo-labels lead PML to misidentify low-margin regions, or whether SKD’s distillation amplifies noise-induced instability.

---

> ### Author Rebuttal · Authors · 2026-03-27
>
> We sincerely thank the reviewer for the **positive assessment** of our paper, especially for recognizing the **optimization-based low-margin analysis**, the **non-trivial design of LDKA**, and the **strong empirical performance with minimal overhead**. We also appreciate the reviewer’s constructive comments on **perturbation design** and **pseudo-label noise**, which help clarify both the current scope of our method. Below we respond to these two points in turn.
>
> **W1[Additional experiment on multi-directional/adaptive perturbations]:** Thank you for this valuable suggestion. Following the reviewer’s suggestion, we conducted supplementary experiments on **Pascal VOC 19-1** with two additional perturbation variants, both of which are used only during training and introduce **no extra trainable modules or inference-time overhead**.
>
> - **Multi-directional perturbation:** in addition to the original normalized gradient direction, we introduce two auxiliary directions that are orthogonalized against it and against each other, and then apply the same propagation-based consistency constraint to the resulting perturbed feature views.
> - **Adaptive perturbation:** we keep the original direction but scale the perturbation magnitude according to the severity of the low-margin condition, so more fragile low-margin samples receive stronger perturbations.
>
> | Method                | 0-19 | 20   | All  |
> |----------------------|-----:|-----:|-----:|
> | Multi-directional    | 83.0 | 69.4 | 82.4 |
> | Adaptive perturbation| **83.2** | 67.0 | 82.4 |
> | Ours                 | 82.9 | **73.9** | **82.5** |
>
> While the two variants slightly improve **old-class** performance, the original single-direction SKD achieves the best **new-class** and **overall** performance. Specifically, our method improves **new-class mIoU by 4.5 points** over the strongest perturbation variant and also achieves the **best overall performance**. This indicates that the submitted design already provides the strongest **stability--plasticity trade-off**.
>
> In addition, following **Reviewer UfuW’s** suggestion, we also conducted further experiments on **adaptive curriculum variants**.
>
> | Method | 0-19 | 20 | All |
> |---|---:|---:|---:|
> | MentorNet | **83.2** | 70.8 | 82.4 |
> | SPL | 83.0 | 70.6 | 82.4 |
> | Ours | 82.9 | **73.9** | **82.5** |
>
> Additional details are provided in our **response to Reviewer UfuW (Q2)**. Those results show that our method achieves the best **new-class** and **overall** performance, further suggesting that the gain of LDKA is not due to a more complicated perturbation or generic adaptive weighting alone, but is more consistently explained by its targeted optimization and stabilization of **low-margin regions**.
>
> **W2[Pseudo-label noise in replay-free CISS]:** Thank you for this important comment. We agree that pseudo-label noise is an inherent issue in **replay-free CISS**. However, this supervision strategy is standard in the setting and is shared by the compared methods; it is **not introduced by LDKA itself**. While some CISS works focus specifically on improving pseudo-label quality, our paper targets a different bottleneck, namely the **optimization difficulty caused by low-margin regions** during incremental learning. Following prior work and for fair comparison, the baseline in our ablation experiments is also built under the same pseudo-label-based setting (p.7, right column, Lines 356–367). Therefore, the observed gains are attributed to the optimization design of LDKA itself, rather than to any change in the supervision protocol.
>
> Regarding whether pseudo-label errors may affect **PML**, pseudo-labels only provide the target index for old-class pixels, whereas the actual low-margin assessment is mainly governed by the **model-derived margin state**, i.e., the competitive logit margin and its associated sensitivity cost, which are further integrated through the robust progress index. The subsequent batch-wise ranking and progress-aware weighting then determine optimization participation. Under this design, PML is less directly dependent on any single pseudo-label assignment.
>
> Similarly, **SKD** does not introduce an additional noisy supervision source. It regularizes **cross-step feature consistency** through curvature damping and perturbation stabilization, with the goal of suppressing unstable responses under small perturbations rather than reinforcing them. **MAD** also operates on class-level competition structure rather than introducing a separate pseudo-label supervision path. Therefore, LDKA is designed on top of the standard pseudo-label regime to **stabilize optimization**, rather than to intensify noise effects.
>
> Once again, we sincerely thank the reviewer for the **careful reading**, the **constructive suggestions**, and the **recognition of the paper’s technical motivation and empirical value**. We appreciate these comments very much, and we have carefully improved the above aspects in the revised manuscript.

---

> > ### Author Rebuttal · Reviewer_f11d · 2026-04-03
> >
> > Thank you for your detailed rebuttal and the additional results provided. They have addressed my concerns, and I maintain my positive recommendation of weak accept.

---

> > > ### Author Response · Authors · 2026-04-03
> > >
> > > Dear Reviewer f11d,
> > >
> > > We would like to express our sincere gratitude for the Reviewer’s recognition of several key strengths of our work, including the rigorous optimization-based analysis of low-margin regions, the identification of high curvature and small stability radius as core bottlenecks in CISS, and the recognition that our design is not a trivial extension, but rather a targeted solution for CISS-specific low-margin optimization. We are also grateful that the Reviewer acknowledged the consistent improvements on both new and old classes, alongside the minimal computational overhead, which supports the practical value of the proposed method.
> > >
> > > Following the Reviewer’s insightful suggestions, we have meticulously incorporated the additional results and clarifications into the revised manuscript.  These revisions include strengthening several explanations in the paper and incorporating the additional experimental evidence discussed during the rebuttal process. Inspired by this positive evaluation, we remain committed to further exploring and advancing this direction in our future research. Thank you once again for the professional and supportive review.
> > >
> > > With sincere appreciation and respect,
> > >
> > > The Authors

---

### Official Review · Reviewer_P9jF · 2026-03-12

**Soundness:** 3
**Presentation:** 3
**Significance:** 3
**Originality:** 3
**Overall Recommendation:** 4
**Confidence:** 3

**Summary:**

This paper studies class-incremental semantic segmentation. The main motivation is that in later incremental steps, many pixels are in low-margin regions, so learning new classes becomes hard and unstable. Based on this, the paper proposes LDKA with three parts: PML for progressive margin-based reweighting, SKD for smoother distillation, and MAD for decoupling confusing classes. Experiments on VOC and ADE20K show good performance, especially on new classes.

**Compliance With Llm Reviewing Policy:**

Affirmed.

**Final Justification:**

The authors’ rebuttal is helpful and has answered most of my main questions in a reasonable way. The clarification on the SKD perturbation term is much better now, and it is clearer that this part is intended to regularize the propagated response after perturbation, not simply the perturbation itself. The explanation of the curvature-related implementation is also more understandable, and the discussion about the scope of MAD is useful. These replies mainly improve the clarity of the method and reduce the ambiguity in my previous concerns, although some wording and formulation details can still be cleaned up in the final version. I keep my original evaluation and score unchanged.

**Key Questions For Authors:**

For the perturbation term in SKD, what is the exact form used in implementation?

- In SKD, are you matching features, logits, or probabilities after perturbation?

- How is the curvature-related term computed in practice? Please clarify the approximation and where it is applied.

- For MAD, is this design mainly for token-based backbones, or is it intended to be more general?

**Limitations:**

The paper has some limitations in clarity and scope. In particular, the SKD part is not fully transparent in formulation, and MAD seems more natural for token-based architectures. I think these limitations should be stated more directly in the paper.

**Strengths And Weaknesses:**

Strengths.
The paper has a clear motivation. The low-margin view is interesting and different from many previous works that mainly focus on forgetting. The method design is also easy to follow, and the experimental results are generally strong.

Weaknesses.

Major issue:
- The formulation of the perturbation loss in SKD is not clear enough. As written, it looks like it may only penalize the perturbation itself, and this does not fully match the text explanation about stability. Since SKD is an important part of the method, this point should be clarified better.
- The curvature damping part lacks enough implementation details. It is unclear how the second-order quantity is computed or approximated in practice, and on which features/tensors it is applied. This makes the method harder to understand and reproduce.

Minor issue:

- Some notation and definitions are not fully consistent. In some places it is not very clear whether the variable means feature, logit, or output representation.

- MAD is based on class-token decoupling, but the scope of this design is not discussed clearly enough. It would be better to state more clearly whether this part mainly targets ViT-style models.

---

> ### Author Rebuttal · Authors · 2026-03-27
>
> We sincerely thank the reviewer for the **positive assessment** of our paper, especially for recognizing the **clear motivation**, the **interesting low-margin perspective**, the **overall method design**, and the **strong experimental performance**. We also appreciate the reviewer’s constructive questions regarding the clarity and scope of **SKD** and **MAD**. Below we clarify these points in a more explicit and implementation-oriented manner.
>
> **W1 & Q1 & Q2 [More details about SKD perturbation stabilization]:** Thank you for these important questions. As described in **Sec. 4.3 (p.5, right column, Lines 229–230)**, the perturbation is injected into the **current-step feature** and then propagated through the network; **Lines 234–237** further state that the responses before and after perturbation are encouraged to remain consistent. In implementation, the perturbation $\delta$ is derived from the gradient of the **distillation discrepancy** $\||z^{t}(x)-z^{t-1}(x)\||_2^2$ with respect to $z^{t}(x)$, while $z^{t-1}(x)$ is produced by the **frozen previous-step model** and treated as fixed. The perturbation is then added to an intermediate current-step feature on the distillation path and propagated through the network to produce the response $\tilde z^{t}(x)$. The consistency term is imposed on the resulting **patch-level feature response after decoder**, i.e., on $z^{t}(x)$ versus $\tilde z^{t}(x)$, rather than on the perturbation magnitude itself. Thus, SKD explicitly matches **decoded features**, not logits or probabilities. We have clarified the notation and description in the revised version so that the perturbation term is understood as enforcing **functional stability of the propagated feature response**, rather than merely shrinking the raw perturbation.
>
> **W2 & W3 & Q3 [Practical computation of curvature damping]:** Thanks for the question. As defined in **Sec. 4.3, Eqs. (16)–(17)**, $h(x)$ formally characterizes the **second-order sensitivity** of the feature-distillation discrepancy $\||z^{t}(x)-z^{t-1}(x)\||_2^2$, establishing our theoretical objective for curvature damping. Here, $z$ denotes the patch-level feature used for cross-step distillation, rather than logits, probabilities, or multi-layer features. In practice, explicitly materializing $\nabla^2$ to compute Eq. (17) is computationally prohibitive. We therefore use an efficient **gradient-penalty surrogate**: we compute the first-order derivative of the discrepancy w.r.t. the current-step feature tensor $z^{t}(x)$ via automatic differentiation (with the computation graph retained), while $z^{t-1}(x)$ is produced by the frozen previous-step model and treated as fixed. By regularizing the squared norm of this gradient, the subsequent backward pass implicitly penalizes second-order variations via **double backpropagation**. This provides a mathematically grounded and efficient surrogate for the theoretical curvature-damping objective. We have revised **Sec. 4.3** to make this distinction between the theoretical formulation and the practical implementation more explicit.
>
> **W4 & Q4 [Generalization across backbones]:** Thank you for this valuable question. MAD is **most naturally instantiated on token-based architectures** in our current design, but the underlying idea is not limited to them. Beyond the ViT results, **Appendix C.3** shows that LDKA also remains effective with a CNN-based **ResNet101** backbone: in ViT-based models, MAD operates on class-specific decoder tokens, while in CNN-based backbones we instantiate the corresponding class representations using the **segmentation classifier weights**. As reported in **Table C.3**, under the same ResNet101 architecture, our method improves mean mIoU over all categories by **0.5** over CoMasTRe (Gong et al., 2024). Following the reviewer’s suggestion, we also added a **SegFormer** comparison, which further shows that our method is **not tied to a specific backbone or decoder form**. On **new classes**, it improves performance by **5.4 points** over the second-best method while maintaining **comparable anti-forgetting ability on old classes**.
>
> | Method | 0-15 | 16-20 | All |
> |---|---:|---:|---:|
> | ILT | 49.1 | 54.0 | 50.3 |
> | MiB | 78.8 | 60.9 | 74.5 |
> | PLOP | 72.5 | 48.4 | 66.8 |
> | SATS | **80.2** | 61.2 | 75.7|
> | Ours |79.7| **66.6** | **76.6** |
>
> Additional details are provided in our response to **Reviewer UfuW (Q3)**. Taken together, these results indicate that our method generalizes across both **Transformer-based** and **CNN-based** backbones.
>
> Once again, we sincerely thank the reviewer for the **careful reading**, the **constructive suggestions**, and the **positive recognition** of the paper’s motivation and results. We fully agree that the presentation of **SKD** and the scope of **MAD** can be made clearer, and we appreciate the opportunity to improve these aspects. We have carefully addressed and improved the above aspects in the revised manuscript.

---

> > ### Author Rebuttal · Reviewer_P9jF · 2026-04-04
> >
> > The rebuttal addressed part of my concerns, but my main concern is still not fully resolved. Therefore, I maintain my score.

---

> > > ### Author Response · Authors · 2026-04-04
> > >
> > > Thanks for your feedback. Since the latest feedback did not specify which particular elements of the main concerns remain unresolved, we would like to further tackle the issue the reviewer raised in the Weakness section in more detail.
> > >
> > > ---
> > >
> > > **W1 [Exact implementation form of the perturbation term in SKD]**
> > >
> > > $L_{\mathrm{pert}}$ is **not** meant to penalize the perturbation itself. Instead, the perturbation is injected into the **intermediate current-step feature on the distillation path**, and the loss constrains the change in the **decoded feature after propagation through the downstream decoder**. Therefore, SKD matches **decoded patch-level features after perturbation**, **not logits** or **probabilities**.
> > >
> > > > **Algorithm 1. Perturbation stabilization in SKD**
> > > >
> > > > **Input:** intermediate current-step feature $f^t(x)$, frozen previous-step decoded feature $z^{t-1}(x)$, downstream decoder $H_\phi$.
> > > >
> > > > **Output:** perturbation-stabilization loss $L_{\mathrm{pert}}$.
> > > >
> > > > **Step 1.** Decode the current-step feature
> > > >
> > > > $
> > > > z^t(x)=H_\phi(f^t(x)).
> > > > $
> > > >
> > > > **Step 2.** Compute the perturbation direction and normalize it
> > > >
> > > > $
> > > > d(x)=\nabla_{f^{t}(x)}\||z^t(x)-z^{t-1}(x)\||_2^2,
> > > > \qquad
> > > > \delta^t(x)=\frac{d(x)}{\||d(x)\||_2}.
> > > > $
> > > >
> > > > **Step 3.** Inject the perturbation into the intermediate feature and propagate it through the decoder
> > > >
> > > > $
> > > > \tilde z^t(x)=H_\phi(f^t(x)+\delta^t(x)).
> > > > $
> > > >
> > > > **Step 4.** Apply perturbation consistency on the decoded feature after propagation
> > > >
> > > > $
> > > > L_{\mathrm{pert}}=E_x[\||z^t(x)-\tilde z^t(x)\||_2^2].
> > > > $
> > > >
> > >
> > > ---
> > >
> > > **W2 [How is the curvature-related term approximated in practice, and where is it applied]**
> > >
> > > Eqs. (16)–(17) define the **theoretical objective**: suppressing the second-order sensitivity of the distillation discrepancy with respect to the **decoded feature used for cross-step distillation**. This quantity is defined on the **feature-distillation path**, not on logits or probabilities.
> > >
> > > To keep the implementation computationally tractable, we do not explicitly materialize $h(x)$ in Eq. (16). Instead, we compute the first-order gradient of the distillation discrepancy with respect to the current-step decoded feature, retain the computation graph, and regularize its squared norm as a practical surrogate for Eq. (17). Backpropagating this surrogate differentiates through the first-order gradient, thereby inducing double backpropagation and providing a practical second-order approximation.
> > >
> > > > **Algorithm 2. Practical computation of curvature damping in SKD**
> > > >
> > > > **Input:** current-step decoded feature $z^t(x)$, frozen previous-step decoded feature $z^{t-1}(x)$.
> > > >
> > > > **Output:** curvature-damping loss $L_{\mathrm{curve}}$.
> > > >
> > > > **Step 1.** Define the distillation discrepancy
> > > >
> > > > $
> > > > \||z^t(x)-z^{t-1}(x)\||_2^2.
> > > > $
> > > >
> > > > **Step 2.** Compute its first-order gradient with respect to the current-step decoded feature, while retaining the computation graph
> > > >
> > > > $
> > > > g(x)=\nabla_{z_{\theta}^{t}(x)}\||z^t(x)-z^{t-1}(x)\||_2^2.
> > > > $
> > > >
> > > > **Step 3.** Regularize the squared gradient norm
> > > >
> > > > $
> > > > L_{\mathrm{curve}}=E_x[\||g(x)\||_2^2].
> > > > $
> > > >
> > > > **Step 4.** Backpropagate the surrogate loss through the retained graph
> > > >
> > > > $
> > > > \nabla_{\theta}L_{\mathrm{curve}} =
> > > > \nabla_{\theta}
> > > > E_x\left[
> > > > \lVert
> > > > \nabla_{z_{\theta}^{t}(x)}
> > > > \||z^t(x)-z^{t-1}(x)\||_2^2
> > > > \rVert_2^2
> > > > \right].
> > > > $
> > > >
> > > > This differentiates through the first-order gradient, thereby inducing **double backpropagation** and providing a practical second-order surrogate.
> > >
> > > ---
> > >
> > > **W3 [Notation and variable definitions]**
> > >
> > > We now explicitly distinguish: $z^t(x)$ as the current-step decoded patch-level feature used for distillation; $z^{t-1}(x)$ as the frozen previous-step decoded feature; $d(x)$ and $\delta^t(x)$ as the perturbation direction and its normalized form; $s(x;\theta)$ as logits; and $p(x;\theta)=\mathrm{softmax}(s(x;\theta))$ as probabilities. This makes clear that both $L_{\mathrm{pert}}$ and $L_{\mathrm{curve}}$ are imposed on the **decoded feature-distillation path**.
> > >
> > > ---
> > >
> > > **W4 [Is MAD primarily designed for ViT-style models, or is it intended to be more general]**
> > >
> > > MAD is **an architecture-agnostic strategy**, not a ViT-only component. What changes across architectures is the **carrier of the class representation**: in **ViT-style models**, MAD operates on class-specific tokens; in **CNN-style backbones**, it is instantiated on the segmentation classifier weights. This is also supported by **Appendix C.3** and the additional cross-backbone results reported in the **first-round rebuttal**. We have revised the manuscript to state this scope more explicitly.
> > >
> > > ---
> > >
> > > We hope this clarification fully addresses your concerns. If any point would benefit from further clarification, we would be very glad to respond more specifically and directly.

---

### Official Review · Reviewer_u5dz · 2026-03-12

**Soundness:** 2
**Presentation:** 1
**Significance:** 3
**Originality:** 3
**Overall Recommendation:** 3
**Confidence:** 3

**Summary:**

The reviewed paper deals with class continual learning in the case of semantic segmentation. The authors identify class confusion between existing classes and new, similar classes as one of the main problems. Also, as the authors do not assume that prior training data is available at the time of the update step, the avoidance of catastrophic forgetting is of importance. To address both problems, the authors design two additional loss terms and a control mechanism that progressively puts more weight to more difficult pixels where the margin of probabilities is low. The two terns consist out of a knowledge distillation term which restricts the difference between features between consecutive update steps. This term includes a standard L^2 - loss term and the norm of the second derivative thereof. In addition, a contrastive loss term is designed to drive apart class feature - called 'token' - similarity between classes that can be confused.

The authors motivate their approach by mathematical fromulae based on the first order Taylor expansion of the class margin and the second derivative of the softmax loss with respect to the difference in logits between the actual class and the next probable one, interpreted as curvature.

In their experiments, the authors perform continual learning experiments on Pascal VOC and ADEK with different amounts of basline classes and additional classes. They compare themselves with 8 recent baselines for ResNet101 and ViT backbones. Further experiments are reported in the appendix. Throughout the experiments the authors find consistent gains of a few % mIoU narrowing the gap to oracle performance considerably.

**Compliance With Llm Reviewing Policy:**

Affirmed.

**Final Justification:**

Despite the fact that the authors suggested improvements in their rebuttal, I still  don't advocate that the paper should be accepted. This is mainly due to the limited theoretical insight that the paper offers, where the theory is based on ad hoc approximations. I will thus keep my score, but due to the numerical experiments, where the paper has its strengths, I do not strongly oppose acceptance.

**Key Questions For Authors:**

What about the computational overhead due to the computation of various second derivatives during training?

Which exact map information is provided during the training update steps? Do the training masks only contain the new masks?

**Limitations:**

No, but not needed

**Strengths And Weaknesses:**

Strength
* The area of continual learning is an established discipline in computer vision. Within this area, the author's methods are broadly applicable.
* The comparison with baselines reveals a strong performance of the proposed method.

Weaknesses
* The mathematical treatment and theoretical motivation given by the paper is convoluted and inconclusive. E.g. the analysis of curvature in the theory affects only the softmax layer (via derivatives w.r.t. the margin), but the distance to the decision boundary is considered in parameter space. the authors somewhat intermingle this.
* The interpretation of one second derivative as general curvature (of decision bounds?) is far from a proper differential geometric computation of curvature. Here the authors overstate their achievements.
* The mathematical derivations based on 1st order Taylor expansion are simple trigonometry and can not be seen as a theoretical contribution. Error correction terms in the first or second order Taylor expansion are dropped as the author please. This is strange as the authors claim an analysis of curvature for their paper.
* I have difficulties recognizing the 'curvature' analysis around eq (3), based on the softmax, with the later trained curvature term (17) based on consecutive features. In particular (3) is in m-space and (17) in parameter space. in as much is the 'theory' then reflected in the actual loss terms?
* I have some problem with Eq. (19)-(21). $\delta_t$ is normalized in the $L^2$-norm by Eq (19). $\tilde z_t$ is then obtained by adding $\delta_t$ to $z_t$ in (20). In (21), we consider the expected $L^2$-norm of the difference between $z_t$ and $\tilde z_t$, hence by (20) the $L^2$-norm of $\delta_t$ being exactly equal to one by (19). Isn't then the expected value in (21) always constant and equal to one and can't participate in backprop? What happened here?
* Although this is an established continual learning problem, having no access to previous training data in many cases is a too restrictive assumption. Therefore the practical relevance of the proposed method is limited.

Minor remarks:
p1:  "New-class performance often shows minimal improvement as the incremental steps becomes longer". I find it hard to connect this fuzzy description to an actual problem. please help with a more concise formulation.
p2 figure 2: Why is the circle with the large stability radius not toutching the decision boundary?
p3, l. 143: $D_t$ badly defined as set with one element
p3, above and below eq (3) $y^*$ and $y$ are both the true class label?
p4, Eq (4) why don't you write down the actual bound 1/4?
p4, below Eq (4) I find the argument of small increments inconclusive as then the gradient and not the second derivative would play the main role.
p4 Eq (7): You write a strict inequality, but this is based on a 1st order Taylor approximation and can be wrong. Therefore, the statement above is not correct either.
p4 eq (8) When the reader sees this, she/he does not yet know the definitions whicj only come afterwards. Add a hint.
p4, Eq (9) define $\kappa$
p4 Eq. (11) Bad layout
p5, Eq  (13) the closing bracket should not be in the exponential
P5 Here z's appear without being clearly defined. Presumably feature maps, but only one layer, many layers or all?
p5, 'Motivation': Give a more concise formulation. Your text is largely allegoric. Check your argument on the instability of the argmax - this does not enter the training, which still can be smooth if the argmax flips, as CE-training is only based on the probabilities.
p5, Eq (16): With respect to which quantity is the gradient taken and how doe the z's dpend on it? Same in Eq (18)
p5, last paragraph, The paper of Dosovitskiyet et al does not contain the word 'class token' - please provide a definition.? Do you mean the averaged features under vision tokens of a given class?
p5 Eq (24): This rather seems to enhance orthogonality than maximizing distance, unless the a's are normalized, in which case that would be equivalent (and go for low cos similarity).
p6, l. 311 - We -> we

---

> ### Author Rebuttal · Authors · 2026-03-27
>
> **W1 [Theoretical motivation]**: The two analyses are coupled through the logit margin: Property 1 characterizes how loss sensitivity increases as the margin shrinks, while Property 2 characterizes how small parameter perturbations can change that margin and reach the decision boundary. They thus capture complementary optimization-related properties of low-margin regions.
>
> **W2 [Term definition]**: Here “curvature” does not mean differential-geometric boundary curvature. In Eq. (3), it denotes the local second-order sensitivity of $L_{\mathrm{seg}}(m)$ w.r.t. the logit margin. Thus, our claim concerns low-margin optimization instability, not boundary geometry.
>
> **W3 [Derivation explanation]**: Property 2 is explicitly a **first-order local approximation**, not an exact derivation; under small perturbations, omitting higher-order remainders is therefore reasonable.
>
> **W4–W5 [Equation explanation]**: Eq. (3) diagnoses why low-margin regions are optimization-sensitive, whereas Eq. (17) implements the corresponding stabilization on the feature-distillation path in SKD. Since logit margins are induced by the underlying feature representation, suppressing excessive second-order sensitivity during cross-step feature alignment serves as a practical feature-space surrogate for mitigating low-margin optimization sensitivity. As stated in **Sec. 4.3 (p.5, right column, Lines 229–231)**, the perturbation is injected into the feature representation and then propagated through the network; thus, Eq. (21) constrains the **propagated response after perturbation**, not the raw difference.
>
> **W6 [Practical value]**: Replay-free CISS is a widely adopted setting. **Recent works [1,2] continue to study replay-free incremental learning, indicating that this regime remains valuable**. Moreover, LDKA also outperforms replay-based baselines such as **SSUL**.
>
> [1] Probabilistic Group Mask Guided Discrete Optimization for Incremental Learning. ICML, 2025.
>
> [2] Pass++: A Dual Bias Reduction Framework for Non-Exemplar Class-Incremental Learning. TPAMI, 2025.
>
> ----
> **P1 & P3.2 [Incremental learning / labels]**: As widely recognized in CISS, learning new classes becomes harder as incremental steps grow. As stated in Sec. 3.1, $y^*$ denotes the pseudo-label for old classes when GT is unavailable, whereas $y$ in Eq. (3) denotes the GT label for new classes.
>
> **P2 [Figure 2 explanation]**: The large radius represents a high-margin region with stronger local perturbation tolerance, i.e., one farther from the decision boundary.
>
> **P4.1–P4.4 [Eqs. (4), (7), (8)]**: The $\tfrac{1}{4}$ bound applies to the intermediate term $\tilde p_y(1-\tilde p_y)$, not directly to $\frac{\partial^2 L_{\mathrm{seg}}}{\partial m^2}$. As shown in **Appendix A.4 (Eqs. 39–43)**, $\frac{\partial^2 L_{\mathrm{seg}}}{\partial m^2}\approx 4\tilde p_y(1-\tilde p_y)\lesssim 1$. Our point below Eq. (4) concerns sensitivity rather than descent direction: while the gradient determines the update direction, the large second derivative in low-margin regions governs sensitivity to small updates. Eq. (7) is stated under the first-order approximation introduced in Eq. (5), and should therefore be interpreted as a local boundary-reaching bound. Under this setting, the inequality is correct. We have also added a note after Eq. (8) clarifying that $L_{\text{seg-PML}}$, $L_{\text{SKD}}$, and $L_{\text{MAD}}$ are introduced in Secs. 4.2–4.4, respectively.
>
> **P4.5 & P5.2–P5.6 [Definitions / interpretations]**: $\kappa$ is already defined before Eq. (9). $z$ denotes the patch-level feature after decoder. The argmax in Eq. (1) only identifies the dominant competing class for margin definition. As shown by Property 1 (Appendix A), the CE loss is continuous, but its second derivative w.r.t. the logit margin peaks around $m \approx 0$; thus, the “instability” refers to extreme local sensitivity in low-margin regions, not to any non-smoothness of the argmax itself. In Eqs. (16) and (18), derivatives are taken w.r.t. $z^t(x)$, while $z^{t-1}(x)$ comes from the frozen previous-step model. “Class token” denotes the class-specific learnable embedding in the head. Eq. (24) suppresses pairwise alignment via $(a_i^\top a_j)^2$; if normalized, this reduces cosine similarity, otherwise it penalizes large pairwise projections.
>
>
> **P3.1 & P5.1 & P6 [Notation/layout issues]**: We have corrected these issues, including rewriting $D_t=\{(x,y^t)\}$ as $D_t=\{(x_n,y_n^t)\}_{n=1}^{N_t}$, fixing the bracket placement in Eq. (13), and correcting “We” to “we”.
>
> ----
> **Q1 [Computation cost]**: We have analyzed in **Table C.4** and **Appendix C.4**. Our method adds only 8.5 ms/iter during training over the second-best method, with negligible inference-time change.
>
> **Q2 [Task setting]**: As clarified in **Sec. 3.1 (Lines 143–146)**, masks at step $t$ contain only the new classes $C_t$; old and future classes are treated as background, and old-class supervision is provided by pseudo-labels.

---

> > ### Author Rebuttal · Reviewer_u5dz · 2026-03-31
> >
> > W1 - my concern remains as the curvatures in parameter and input space remain to be not separated. That both imply second derivatives of the softmax does not resolve this concern, this was clear from the beginning. In their rebuttal the authors now call these two quantities complementary, but there is no amandment of the text announced.
> > W2 - my concern is not resolved as the authors speak of curvature which has a well defined mathematical meaning tied to differential geometry. If they would like to speak about second derivatives of some functions in specific directions, the authors should say so. Again, no amandment of the text is announced.
> > W3 - This is agreed, but my criticism applied to this non controlled approximation.
> > W4-5 - If $\delta^t$ in eq (19) and $z^t$, $\tilde{z}^t$ (20) is not the same as in (21), the authors should make this clear via notation. As it stands, (21) is constant =1, which does not make sense (follow the simple argument in my review). The explanation given that features somehow pass trough a network is not explaining what happens here. the authors do not seem to see the necessity to change.
> >
> > I see the minor remarks as mostly resolved (they did not enter the score). Thank you for the runtime analysis - acknowledged.

---

> > > ### Author Response · Authors · 2026-04-01
> > >
> > > We sincerely appreciate the attention given to our work. We recognize that the remaining concerns primarily stem from **highly condensed terminology** (W1/W2/W3) and **overloaded notation** (W4/W5). To remove this ambiguity, we **have made the corresponding amendments explicit in the revised manuscript and Appendix**.
> > >
> > > ---
> > >
> > > **W1/W2.** We do **not** treat the quantities in margin, parameter, and representation space as the same object. Rather, they are **diagnoses** that jointly motivate **Secs. 4.2–4.4**.
> > >
> > > **Property 1.** Eqs. (3)–(4) characterize the **directional second-order sensitivity** of $L_{\mathrm{seg}}$ with respect to the logit margin $m$. As shown in **Appendix A (Eqs. (39)–(43))**, this quantity peaks when $m \approx 0$, identifying low-margin regions as highly sensitive.
> > >
> > > **Property 2.** Eqs. (5)–(7), together with **Appendix B (Eqs. (47)–(54))**, characterize the **local stability radius** $r$, i.e., the smallest parameter perturbation required to flip the sign of the margin under a first-order local approximation. Thus, low-margin regions are also locally fragile under incremental updates.
> > >
> > > These two diagnoses jointly motivate **Secs. 4.2–4.4**: PML shifts optimization from high-margin, more stable regions to low-margin, more fragile ones; SKD regularizes the downstream response against intermediate perturbations; and MAD decouples highly competitive class pairs, since sensitivity and fragility in low-margin regions readily amplify class confusion.
> > >
> > > Accordingly, the revised text now explicitly separates margin space, parameter space, and method-level effects, and replaces the broad term *curvature*, where appropriate, with the more precise expression **directional second-order sensitivity along the margin direction**.
> > >
> > > To make the **bridge** between the margin-based diagnosis and the representation-level stabilizer explicit, and building on **Appendix A (especially Eqs. (39)–(43))**, we have added an Appendix clarification based on the **Hessian chain rule**:
> > >
> > > $ \nabla_f^2 L_{\mathrm{seg}} = \frac{\partial^2 L_{\mathrm{seg}}}{\partial m^2}(\nabla_f m)(\nabla_f m)^\top + \frac{\partial L_{\mathrm{seg}}}{\partial m}\ \nabla_f^2 m. $
> > >
> > > This makes the relation explicit: the margin-based second-order sensitivity in Eq. (3) propagates to the representation level through the margin.
> > >
> > > Consistent with **Appendix B (Eqs. (47)–(54))**,
> > >
> > > $ \qquad r(x;\theta) \approx \frac{|m(x;\theta)|}{\||\nabla_\theta m(x;\theta)\||}. $
> > >
> > > This makes the relation explicit: in low-margin regions, small incremental parameter perturbations can induce larger relative margin changes, resulting in a smaller local stability radius. We have **enhanced the above presentation** in the revised version.
> > >
> > > ---
> > >
> > > **W3.** Following the reviewer’s suggestions, we have **refined the presentation** in the revised version. Property 2 is used as a locally controlled approximation in the **small-perturbation regime** relevant to incremental updates. More specifically, for sufficiently small perturbations $\Delta\theta$, the Taylor expansion is written as
> > >
> > > $ m(x;\theta+\Delta\theta) = m(x;\theta) + \nabla_\theta m(x;\theta)^\top \Delta\theta + O(\||\Delta\theta\||^2). $
> > >
> > >
> > > ---
> > >
> > > **W4–W5.** In alignment with the reviewer’s suggestions, we have **formalized the technical exposition** in the revised manuscript to **eliminate any remaining ambiguity**. Under a literal reading of the previous compressed notation, if Eq. (20) were interpreted as
> > >
> > > $ \tilde z^t = z^t + \delta^t, $
> > >
> > > **then Eq. (21) could be misread** as
> > >
> > > $ \mathbb{E}\big[\||z^t-\tilde z^t\||_2^2\big] = \mathbb{E}\big[\||\delta^t\||_2^2\big]. $
> > >
> > > This becomes constant after the normalization in Eq. (19). This is exactly the reading reflected in the review.
> > >
> > > **However, this is not the intended formulation.** Here, $f^t$ denotes an intermediate feature representation, $z^t = H_\phi(f^t)$ denotes the downstream propagated response being regularized, and $H_\phi$ denotes the decoder mapping after $f^t$. The intended formulation is
> > >
> > > $ z^t = H_\phi(f^t), \qquad \tilde z^t = H_\phi(f^t+\delta^t), $
> > >
> > > and therefore
> > >
> > > $ L_{\mathrm{pert}} = \mathbb{E}[\||H_\phi(f^t)-H_\phi(f^t+\delta^t)\||_2^2]. $
> > >
> > > Under this clarified formulation, **the term is not a constant-norm identity**; rather, it constrains the **functional response after perturbation**.  We have therefore updated the notation and explanation around Eqs. (20)–(21) so that the perturbation target and the propagated response are no longer overloaded by the same symbol.
> > >
> > > ---
> > >
> > > **We hope that these further technical formalizations and explanations, made in direct response to the reviewer’s suggestions, will meet with the reviewer’s approval. We thank the reviewer for the engagement with our work.**

---

### Decision · Program_Chairs · 2026-04-30

**Decision:**

Accept (regular)

**Comment:**

Dear authors,

This draft received overall positive rating (5, 3, 4, 4). Reviewer u5dz assigned rating 3 and indicated that paper offers limited theoretical insight, but  also did not oppose acceptance.

Please update draft per reviewer comments and rebuttal for the camera ready submission.

Congratulations.

Regards

AC